# Spontaneous electric-polarization topology in confined ferroelectric nematics

Jidan Yang[1,3], Yu Zou[1,3], Wentao Tang[1,3], Jinxing Li[1], Mingjun Huang ®[1,2] ✉ & Satoshi Aya ®[1,2] ✉

Topological textures have fascinated people in different areas of physics and technologies. However, the observations are limited in magnetic and solid-state ferroelectric systems. Ferroelectric nematic is the first liquid-state ferroelectric that would carry many possibilities of spatially-distributed polarization fields. Contrary to traditional magnetic or crystalline systems, anisotropic liquid crystal interactions can compete with the polarization counterparts, thereby setting a challenge in understating their interplays and the resultant topologies. Here, we discover chiral polarization meron-like structures, which appear during the emergence and growth of quasi-2D ferroelectric nematic domains. The chirality can emerge spontaneously in polar textures and can be additionally biased by introducing chiral dopants. Such micrometre-scale polarization textures are the modified electric variants of the magnetic merons. Both experimental and an extended mean-field modelling reveal that the polarization strength plays a dedicated role in determining polarization topology, providing a guide for exploring diverse polar textures in strongly-polarized liquid crystals.

Topological defects arise when symmetry is broken. The complexity and diversity of the resulting spin or magnetic textures, such as skyrmions, merons, and the like, have attracted much attention for centuries[1–12]. In recent decades, it has been demonstrated that such vectorized fields with quasi-particle nature serve as an effective approach for integrating and storing magnetic, electronic, optical, and quantum information[13–19]. The findings trigger new challenges for the exploration of the unidentified topological fields, and their designability and controllability.

Quasi-particle topologies with vectorized fields in the context of (quasi-)2D systems, including rich species of vortices, skyrmions, and merons, are usually characterized by three topological numbers: the Pontryagin (or skyrmion) number, vortex number, and helical number. The Pontryagin number, $Q$, defines how many times a unit vector that defines the average orientation, the so-called director $\mathbf{n}$, wraps the unit sphere[4,20,21], represented by

$$Q = \frac{1}{4\pi} \int \mathbf{n} \cdot (\partial_x \mathbf{n} \times \partial_y \mathbf{n}) \mathrm{d}x \mathrm{d}y. \tag{1}$$

$x$ and $y$ are the Cartesian coordinates. However, the usage of only $Q$ cannot fully describe various degenerated topological states. For example, the skyrmion and antiskyrmion cannot be distinguished only by $Q$. Thus, the additional vortex and helical numbers, $Q_v$ and $Q_h$, should be introduced to account for in-plane rotation and relative phase variation[4,22]. By using these parameters, vectorized fields can be expressed

$$\begin{aligned}\mathbf{n}(\theta, \phi) &= (\sin\theta \cos\phi, \sin\theta \sin\phi, \cos\theta) \\ &= (\sin\theta \cos(Q_v\varphi + Q_h), \sin\theta \sin(Q_v\varphi + Q_h), \cos\theta).\end{aligned} \tag{2}$$

[1]South China Advanced Institute for Soft Matter Science and Technology (AISMST), School of Emergent Soft Matter, South China University of Technology, Guangzhou 510640, China. [2]Guangdong Provincial Key Laboratory of Functional and Intelligent Hybrid Materials and Devices, South China University of Technology, Guangzhou 510640, China. [3]These authors contributed equally: Jidan Yang, Yu Zou, Wentao Tang. ✉e-mail: huangmj25@scut.edu.cn; satoshiaya@scut.edu.cn

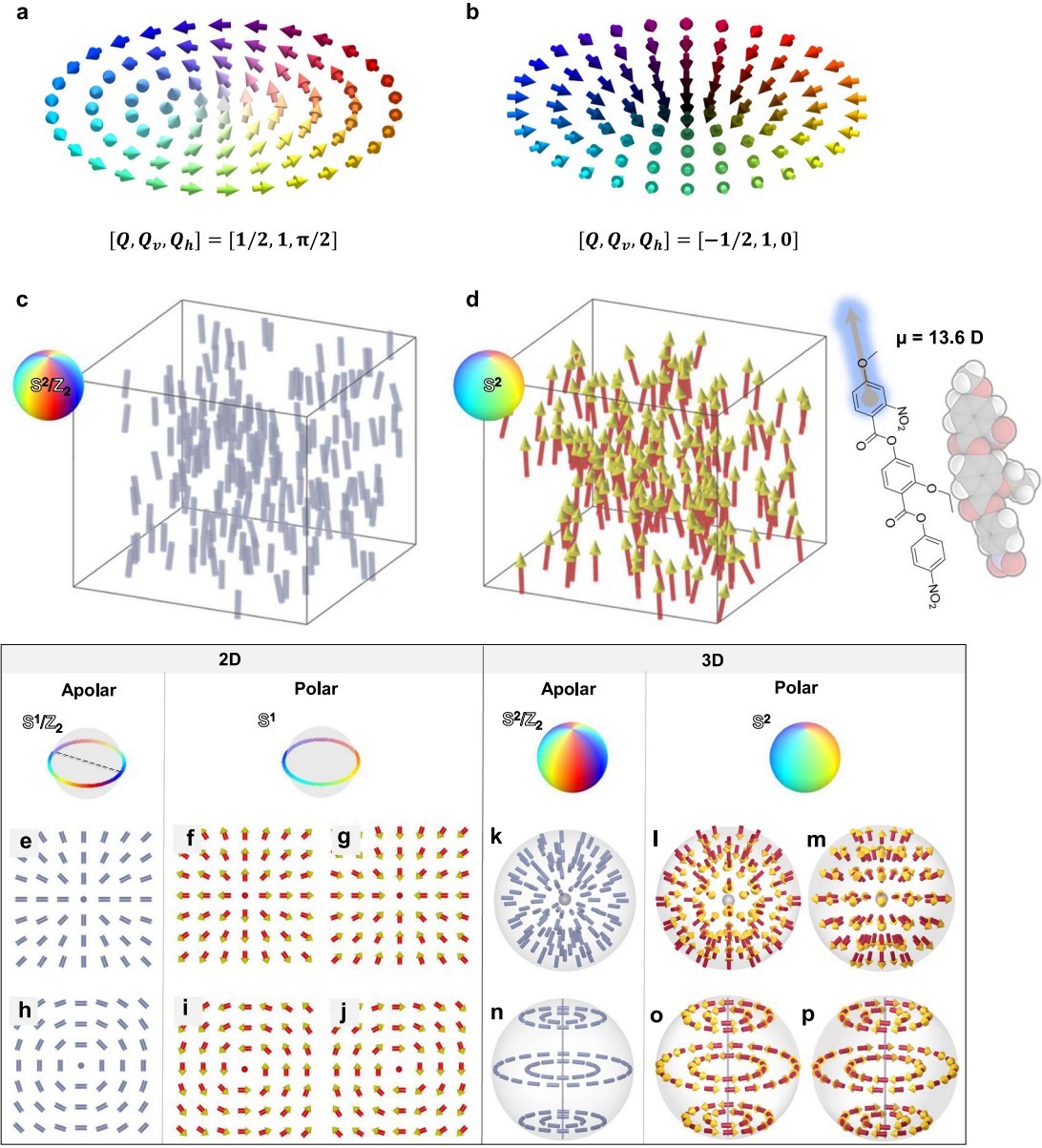

**Fig. 1 | Topological analogy between nonpolar and polar systems.**
**a**, **b** Illustrations of 2D merons with different sets of topological charge, vorticity number and helicity number, i.e., $[Q, Q_v, Q_h]$. **a** Bloch-type meron with $[Q, Q_v, Q_h] = [1/2, 1, \pi/2]$. **b** Néel-type anti-meron with $[Q, Q_v, Q_h] = [-1/2, 1, 0]$. **c** Head-to-tail-equivalent director field of nonpolar nematics. **d** Head-to-tail-inequivalent polarization field of ferroelectric nematics. The polarization entity of RM-OC$_2$ is shown. **e** A +1 hedgehog-type defect in 2D nonpolar nematics. **f**, **g** The polar counterparts of divergent (**f**) and convergent (**g**) polarization fields of (**e**). **h** A concentric-type defect in 2D nonpolar nematics. **i**, **j** The polar counterparts of right hand (**i**) and left hand (**j**) polarization fields of (**h**). **k** A +1 hedgehog-type defect in 3D nonpolar nematics. **l**, **m** The polar counterparts of divergent (**l**) and convergent (**m**) polarization fields of (**k**). **n** A concentric-type defect in 3D nonpolar nematics. **o**, **p** The trivial polar counterparts of right hand (**o**) and left hand (**p**) polarization fields of (**n**).

where $\theta$ and $\phi$ are polar and azimuthal angles of director **n**, and $\varphi$ is the angle part of the polar coordinates for expressing the Cartesian coordinate $(x, y)$. For example, while an anticlockwise Bloch-type meron with an upward vector in the core exhibits a combination of the topological numbers $[Q, Q_v, Q_h] = [1/2, 1, \pi/2]$ (Fig. 1a), a divergent anti-meron with a downward vector in the core shows $[Q, Q_v, Q_h] = [-1/2, 1, 0]$ (Fig. 1b). Though such topological textures have been widely found in magnetic systems, it is not until recently that several electric analogs were reported in multiferroic crystalline systems with superlattices[23-26]. This raises fundamental questions of whether more diverse polarization topological textures, beyond the magnetic counterparts, can emerge in broader physical systems, and how an additional interaction would bring unknown topologies into existence.

The recent discovery of a fluidic nematic-state ferroelectric, the so-called $N_F$ state[27-31], is the first instance of liquid-matter ferroelectric with high fluidity. Due to the liquid crystalline (LC) nature, their inherent elasticity and polarity, and multifaceted interactions with confined surfaces and responsivity to external fields, offer us fertile ground for exploring exotic states and emergent phenomena in soft matter physics. Therein, the polarization field is directly envisioned by second harmonic generation (SHG) microscopies thanks to its high nonlinear coefficient[32-39], and the system is easy to handle and manipulate. The advantages of the material category extend beyond the traditional magnetic systems and solid-state ferroelectrics if interplays between polarity and LC orientation happen. Here, we report on the discovery of various nontrivial polarization meron variants in the $N_F$ state. They exhibit spontaneous chirality symmetry

breaking regardless of the achiral nature of the polar entities. By incorporating dipolar interaction, we extend the n-director-based mean-field theory with the Frank-Oseen potential[40–42] for describing the observed polarization topologies. The findings establish the fundamental physics of soft matter polarization textures and permit the exploration of the parameter space for their stabilization and diversification.

## Results

### Generation of quasi-2D polarization texture in $N_F$ droplets

Traditional nematic (N) LCs that possess only the orientational order are apolar. The local average orientation of molecules is denoted by the director **n**. It exhibits the head-to-tail invariant nature in the apolar nematics, thereby leading to $\mathbf{n} \equiv -\mathbf{n}$ in the order parameter space of $\mathbf{S}^2/\mathbf{Z}_2$ in 3D (Fig. 1c)[43,44]. This means the order parameter of the apolar nematic is a second-rank tensor. The $N_F$ state is the polar counterpart of the traditional nematics, which imposes an additional orientational requirement: $\mathbf{n} - \mathbf{n}$ in the order parameter space of $\mathbf{S}^2$ where the head and tail of the director are differentiable (Fig. 1d)[4,20,45,46]. This polar nature of the orientational directionality is expected to give much more diverse topologies. As seen in the following, even limiting to trivial situations, the emergence of the vectorized polarization at least doubles the variety of the topology compared with the apolar N state. As some explicit examples, we consider correspondence between some topological structures in the apolar and polar systems for both 2D and 3D cases (Fig. 1e–p). In 2D case, the apolar and polar counterparts stay in the order parameter spaces of $\mathbf{S}^1/\mathbf{Z}_2$ and $\mathbf{S}^1$, respectively. The topological defects in the two types of director fields are points in the cores for both the apolar and polar systems (Fig. 1e–j). In 3D case under a spherical confinement, the order parameter spaces become $\mathbf{S}^2/\mathbf{Z}_2$ and $\mathbf{S}^2$. Extending the 2D radial director fields (Fig. 1e) filling the sphere, the +1 radial defects can be trivially transformed to hedgehog-type defects under the homeotropic boundary condition (Fig. 1k). In the polar versions, the hedgehog branches into two distinct states that correspond to the radial divergent and convergent polarization fields (Fig. 1l, m). On the other hand, the 3D counterparts of the 2D concentric +1 defects under the planar boundary condition remain diverse. The direct space-filling of the concentric +1 defects in 3D sphere can possibly carry line disclination (Fig. 1n–p) or be nontrivial. The topological complexity will be affected by the competition of elasticity and polar interactions as discussed below.

In this study, we synthesized a new $N_F$ material with a large dipole moment of ~13.6 D, named RM-OC$_2$ (Fig. 1d and Supplementary Fig. 1a). The structure is based on RM734[29,47] with additions of a nitro and an ethoxy group at the third and second benzene rings, respectively. Such a molecular design leads to a larger lateral bulkiness and a more oblique molecular dipole with respect to the molecular long axis. Since the molecular shape significantly deviates from the ideal rod, we anticipated the traditional nematic state is not preferred. Instead, anisotropic polar interactions serve as the main driving force for generating an anisotropic orientation and thus stabilizing a polarity-driven nematic phase, i.e., the $N_F$ state. Fitting to our expectation, though most of the previously-reported $N_F$ states emerge from the high-temperature apolar N state[27,29,48,49], RM-OC$_2$ exhibits a direct transition from the isotropic liquid (Iso) to the $N_F$ state upon cooling with the phase sequence of Iso-[64.6 °C]-$N_F$ (Fig. 2a–d). On the heating, the material shows several recrystallization processes i.e., $N_F$-[67.1 °C]-Iso-[82.1 °C]-unknown crystal X1-[105–120 °C]-unknown crystal X2-[140.1 °C]-Iso on heating (Supplementary Figs. 1b–j). The recrystallization processes suggest the Iso phase is metastable with respect to some crystalline forms in the range of 80–150 °C. Like the other emerging $N_F$ materials[48–50], both the dielectric constant and second harmonic generation signal dramatically increase upon transitioning into the $N_F$ state (Fig. 2a). The

direct Iso-$N_F$ transition pathway modifies the $N_F$ topology because of the evolution of the polarization field from disorder states. In the previously reported cases with the N-$N_F$ transition[27,29,48–50], the low-temperature $N_F$ polarization (or director) field develops by breaking the head-to-tail invariance of **n** of the high-temperature apolar N state. It accompanies a collective flipping process of the molecular polarity to a preferred direction over a length scale larger than micron-scale[27,47,51]. Therefore, the orientational direction of the molecular polarity is affected by the orientational pattern of the high-temperature apolar N state. Upon the direct Iso-$N_F$ transition, the polar 'spherulite' domains appear from the Iso background through nucleation and growth processes (Fig. 2b, c), finalizing into the band-like texture with disclination lines (Fig. 2d and Supplementary Figs. 2, 3). Such a phase evolutional pathway allows us to track the pure emerging polarization field from a disordered state.

### Observation of polarization meron variants

In $N_F$ droplets, the texture (Fig. 3a–c and Supplementary Fig. 4) cannot be brought into an extinction status under the crossed polarizers, especially in the center. Similar polarized light microscopy (PLM) images were also found recently in some other $N_F$ systems[50,52,53]. These observations suggest the $N_F$ droplets carry a spontaneous twist along the depth direction, which causes the optical rotation effects. This was immediately confirmed in detail by PLM decrossing technique. The clockwise (Fig. 3a) or counter-clockwise (Fig. 3c) rotation of the analyzer distinguishes two different types of textures: at specific decrossing conditions, one type turns to a dark state (i.e., extinction) and the other remains bright. This indicates the two types have opposite handedness. With the insertion of a quarter-wave plate, it seems that the two types of patterns correspond to concentric director fields with left hand (Fig. 3d, e) and right hand (Fig. 3f, g) spirals. To obtain 3D orientational information, we employ the fluorescence confocal polarizing microscopy (FCPM) and deduce the director field[40,54] ("Methods"). In principle, the polarizing fluorescence process gives rise to a $\cos^2\beta$ orientational dependence of the fluorescence signal, where β is the angle between the incident linear polarization and local director. Also, the out-of-plane of director tilting causes the decrease of fluorescence. Thus, analyzing the spatial distribution of fluorescence intensity at proper combination of polarization conditions allows us to calculate the director field ("Methods"). Technically, while the linear polarization pumping makes the in-plane and out-of-plane director tilting undifferentiable, the addition of experiment with a circular polarized pumping light at the same field-of-view extracts the pure information of the out-of-plane tilting[54]. Therefore, the in-plane and out-of-plane director tilting information are uncoupled and accessible. Worth noting, it is exclusively possible to know the average molecular orientation, but the polarization information is still missing. Figure 3h–k demonstrates the FCPM images of the two types of patterns. It confirms that the $N_F$ droplets confined in the thin LC cells show cylindrical shape with some curvatures. From the 2D cross sections at various depths (Supplementary Fig. 5), it is seen that, consistent with the PLM observation, two types of director fields spontaneously twist like spirals. The 3D orientational information for deriving the overall topology can be further deduced by calculating the 3D distribution of director orientation. As seen from the FCPM under a circular polarized pumping light, a decrease of fluorescence occurs near the droplet center, while the other areas show a constant fluorescence intensity (Supplementary Fig. 6c). In the depth cross section profile, the fluorescence profile in the droplet center appears as a broaden black cylinder along the surface normal. This suggests the director field exhibits either a significant out-of-plane tilting towards the center (i.e., homeotropic-like in the center) or a disclination line. The possibility of the existence of a line disclination is excluded by experiment. The fluorescence profile of a designed line disclination by photo-pattern is thin and sharp (Supplementary Fig. 6a), which disagrees with the

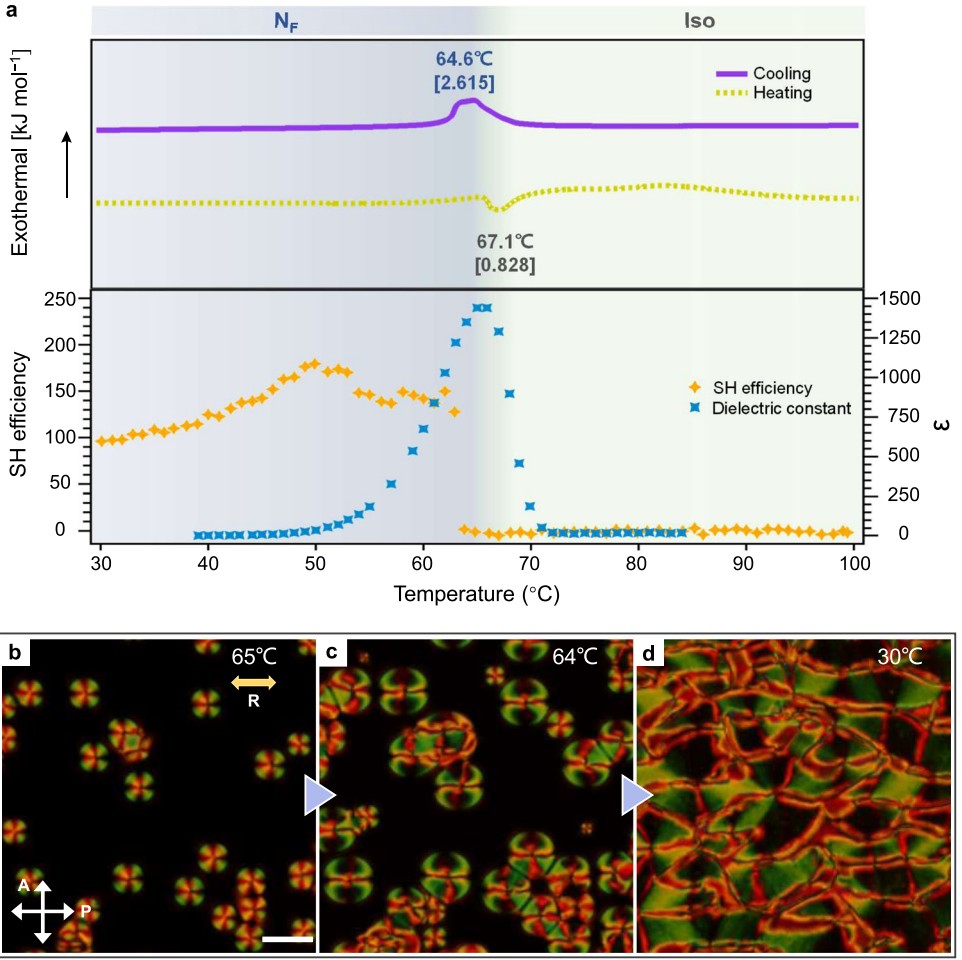

**Fig. 2 | General properties of the ferroelectric nematic. a** Top chart shows DSC curves of RM-OC$_2$ during cooling (violet solid line) and heating (yellow dash line) (rate: 3.0 K min$^{-1}$). Enthalpies of the corresponding phase transitions are shown in the square brackets [kJ mol$^{-1}$]. Bottom chart shows the temperature dependencies of the apparent dielectric constant at a frequency of 25 Hz (blue squares) and SH signal (yellow rhombuses). SH signal is defined as the SH signal intensity ratio of samples to that of a reference Y-cut quartz plate. **b–d** The PLM texture evolution of RM-OC$_2$ during cooling. Optical graphs are taken within a planar cell. Rubbing direction is indicated by the yellow arrow. Scale bar: 50 µm.

broaden black cylinder in the fluorescence profile of droplets (Supplementary Fig. 6b). This suggests that the broad range of light extinction corresponds to the out-of-plane tilting of director. Therefore, the observed topology is limited to either a meron-like structure or an escaped vortex structure as further discussed later in "Discussion". Additional simulations of the fluorescence profile of these two possible structures deduced from numerical structural relaxation ("Methods", Discussion and Supplementary Figs. 7a–c) further confirm the topology of the droplet. It is clear that the signal profile of meron-like structures (Supplementary Fig. 7b) is consistent with our experimental result (Supplementary Fig. 7d), suggesting the observed topology is meron-like structures (Fig. 3l, m) if such a polarization field exists in addition to the orientational field.

Unsimilar to the traditional head-to-tail-equivalent director field, the polarization field should be additionally addressed by SHG microscopy methods to investigate the head-to-tail-inequivalent polarization field[36,48]. Figure 4a demonstrates a large-area SHG confocal polarizing microscopy (SHG-CPM) image of the emerging N$_F$ droplets in the cell midplane[55]. The large contrast between the N$_F$ droplets and apolar background signals visualizes the polarization field. Distinct to the polarizing fluorescence process with a cos$^2$β orientational dependence, the SHG process exhibits a cos$^4$β orientational dependence of the SH signal. The SH signal is the largest/smallest when the polarization of

excitation light is (anti)parallel/perpendicular to the local polarization. It is clearly seen that the coexistence of the two types of director fields as deduced from FCPM emits a strong SH signal. Worth noting, the signal distributions in FCPM and SHG images are well consistent, suggesting the local polarization is parallel to the director (Figs. 3h, j and 4b–g). In the normal SHG measurement, while the existence of the polar symmetry is probed, the head and tail of the polarization cannot be differentiated. It means that the information on the vectorized orientation of the polarization is overlooked. SHG interferometry (SHG-I) microscopy is an extended SHG microscopy technique that 'sees' polarization vector by differentiating the relative difference of the optical phase information[33,36,56] ("Methods"). Figure 4h–k confirms four types of distinct polarized SHG-I images. The four types of polarization meron-like structures could not be resolved by either FCPM or SHG-CPM where only two types of patterns have been seen (Figs. 3h–k and 4a). They correspond to four combinations of left or right hand meron-like structures with divergent or convergent polarization fields (Fig. 4p–s). The chiral symmetry breaking under the confinement occurs due to the delicate balance between the elastic and polar interactions as explained in "Discussion". They show almost equal number of appearances (inset in Fig. 4a), indicating that the handedness of the spontaneous chirality in the polarization meron-like structures is induced by accident.

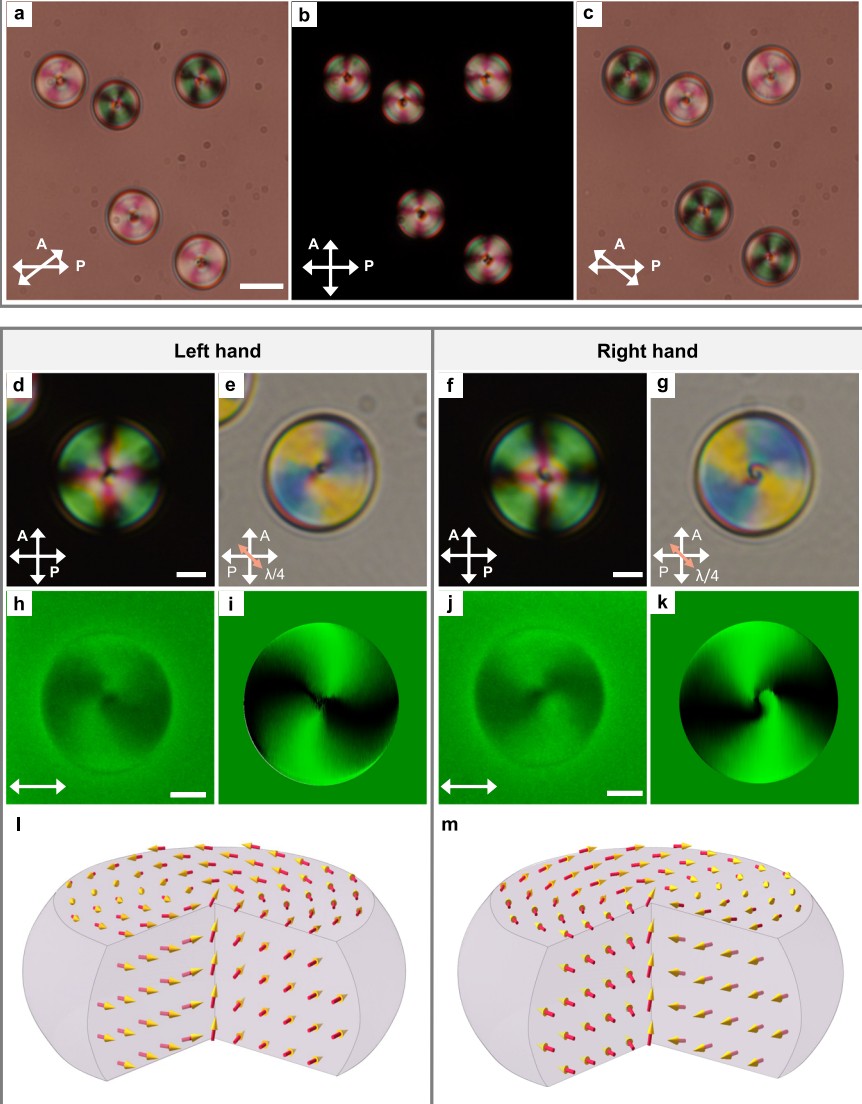

**Fig. 3 | N_F droplets with opposite handedness. a–c** PLM images of N_F droplets under different polarizer conditions. The analyzer is rotated 52° clockwise (**a**); the analyzer is perpendicular to the polarizer (**b**); the analyzer is rotated 52° anticlockwise (**c**). Scale bar, 20 μm. **d–g** PLM images of left hand (**d, e**) and right hand (**f, g**) N_F droplets under crossed polarizers without (**d, f**), and with a quarter-wave plate (**e, g**). The angle between the fast axis of the quarter-wave plate and the polarizer is 45°. The orange arrow indicates the fast axis of the quarter-wave plate. Scale bars, 5 μm. **h–k** XY-cross-sectional FCPM images in the midplane of the cell visualized by a linearly polarization **P**. Scale bars, 5 μm. The white arrows represent the linear polarization as the incident light. The images of (**h, j**) were obtained experimentally and images of (**i, k**) are the numerical results. **l, m** The calculation results were based on the reconstructed director fields obtained from FCPM data.

## Chirality biasing of meron invariants

Introducing chiral agents is a general way for producing an additional twist and biasing the handedness. Here, for retaining the same topological structures, we add a very small amount of two commercial chiral dopants, right-handed R811 or left-handed S811, into the system to see the degree of the symmetry breaking. As shown in Supplementary Figure 8, above a threshold weight percentage, >0.1 wt%, the handedness of the system is totally biased to that of the chiral dopants. Combining PLM, SHG-CPM and CFPM, the system exhibits only two types of the polarization meron-like structures (Fig. 5 and Supplementary Fig. 9), surviving from the original four types of polarization meron-like structures (Fig. 4p–s). The polarization fields are again clarified by SHG-I images: convergent or divergent polarization field for a certain handedness of twist determined by the chiral dopant (Supplementary Fig. 10). These observations verify that the system stabilizes one of the twisted handedness under the action of the chiral dopants.

## Discussion

### Effect of polar interactions on polarization topology

We discuss what leads to the spontaneous chirality in the achiral N_F system. Due to the emergent polarization field, polar interactions as well as the traditional elastic contributions are vital for dedicating the global topology. Therefore, we consider an extended Frank-Oseen free energy function of the system by incorporating dipolar interactions besides the Landau, elastic, surface anchoring, and defect core energies ("Methods"). We uncover five possible polarization states in the curved cylindrically confined space: polarization vortex (Fig. 6a), escaped vortex (Fig. 6b), concentric meron-like structure (C-meron; Fig. 6c), divergent meron-like structures (D-meron, including the convergent, so negatively divergent, meron-like structure; Fig. 6d) and bipolar structure (Fig. 6e). In the simplest polarization vortex (Fig. 6a), the polarization field arranges in a pure concentric manner and lies on the sample plane (i.e., wrapping the equator of the order parameter space **S**² sphere). This leads to a disclination line in the center. In the

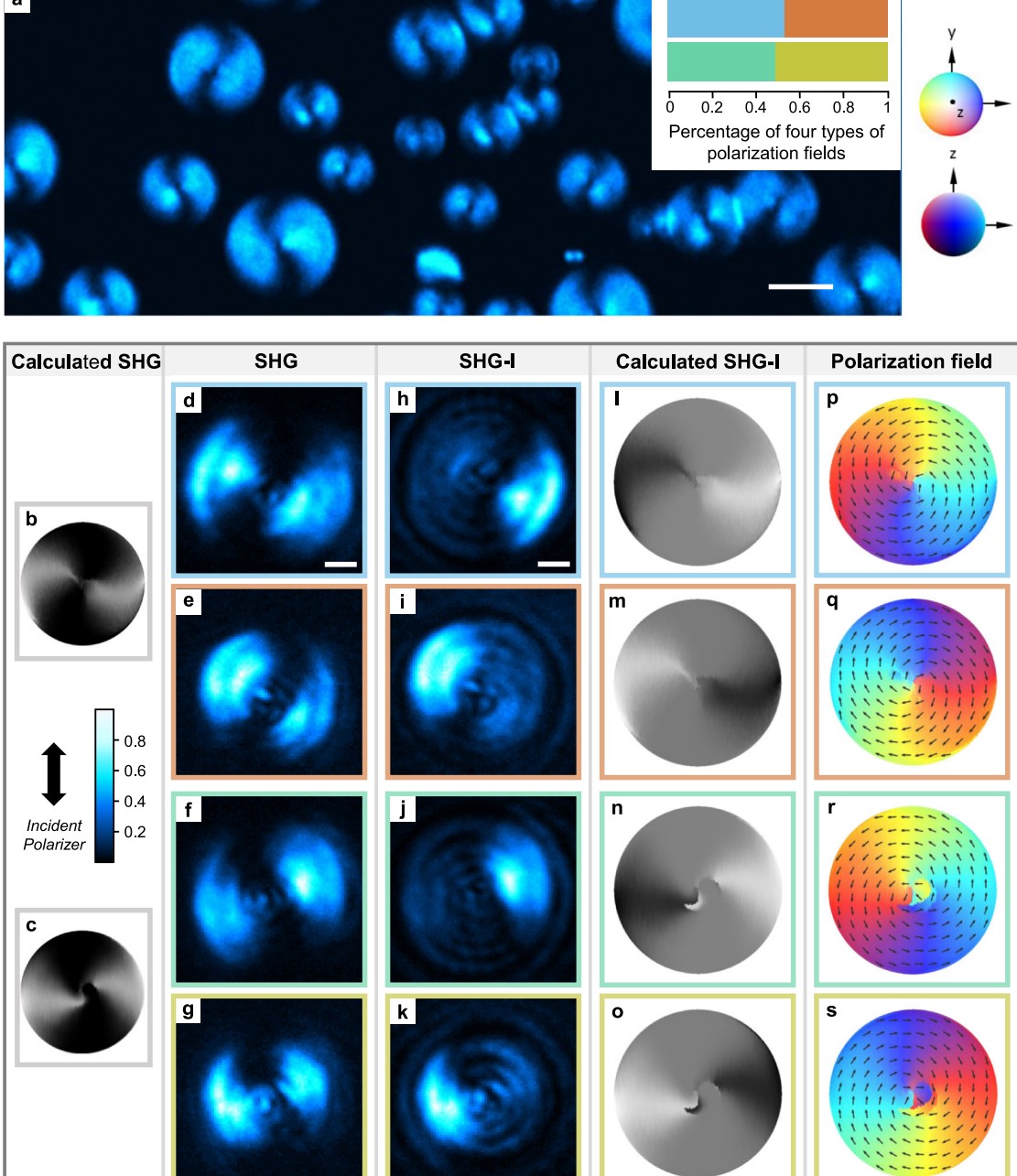

**Fig. 4 | Polarization textures probed by SHG and SHG-I microscopy. a** XY-cross-sectional SHG-CPM images of the $N_F$ droplets. The polarization of the incident laser is shown as the black double arrow. Scale bar, 10 μm. The inset shows the statistic number counts of four types of polarization fields observed in the experiment. **d–s** Four types of the polarization fields **P(r)** are demonstrated in rectangles with blue, orange, green and yellow edges respectively. Experimental SHG microscopy (**d–g**) and SHG-I microscopy (**h–k**) observations of 4 types of polarization textures. In SHG microscopy observation, two types of polarization fields with the same chirality (**d–g**) are degenerate. In SHG-I microscopy observations (**h–k**), the interference conditions are the same. Scale bars, 5 μm. The signal reaches its maximum when polarization direction of the fundamental light is parallel to the electric polarization. **b, c** Calculated SHG intensity obtained from fitted director fields **n(r)**. The director fields are obtained from FCPM data. **l–o** Calculated SHG-I intensity obtain from fitted polarization fields **P(r)**. **p–s** Reconstructed polarization fields with right-handed convergent polarization field (**p**), left-handed divergent polarization field (**q**), right-handed divergent polarization field (**r**), and left-handed convergent polarization field (**s**).

escaped vortex (Fig. 6b), the polarization continuously flips along the depth direction, enabling the disclination line of simple polarization vortex to escape onto the middle plane. Now, the polarizations span over the whole $S^2$ sphere. Since the disclination line of the simple polarization vortex costs much larger energy than that of the point defect of the escaped vortex in our experimental geometry (see "Methods" for free energy descriptions), the escaped vortex is more stable than the simple polarization vortex in most of the simulation conditions (Fig. 6g and Supplementary Figs. 11–13). Meron-like structures are distinct from the vortices because they do not exhibit singularities. C-meron demonstrates a pure concentric polarization field except for the core region (Fig. 6c). D-meron exhibits additional directional splay-bend distortions (Fig. 6d), which is consistent with the experimental observation of the in-plane handedness of meron-like structures. The bipolar structure possesses two boojum defects on the north and south poles (Fig. 6e), which has been well studied in the

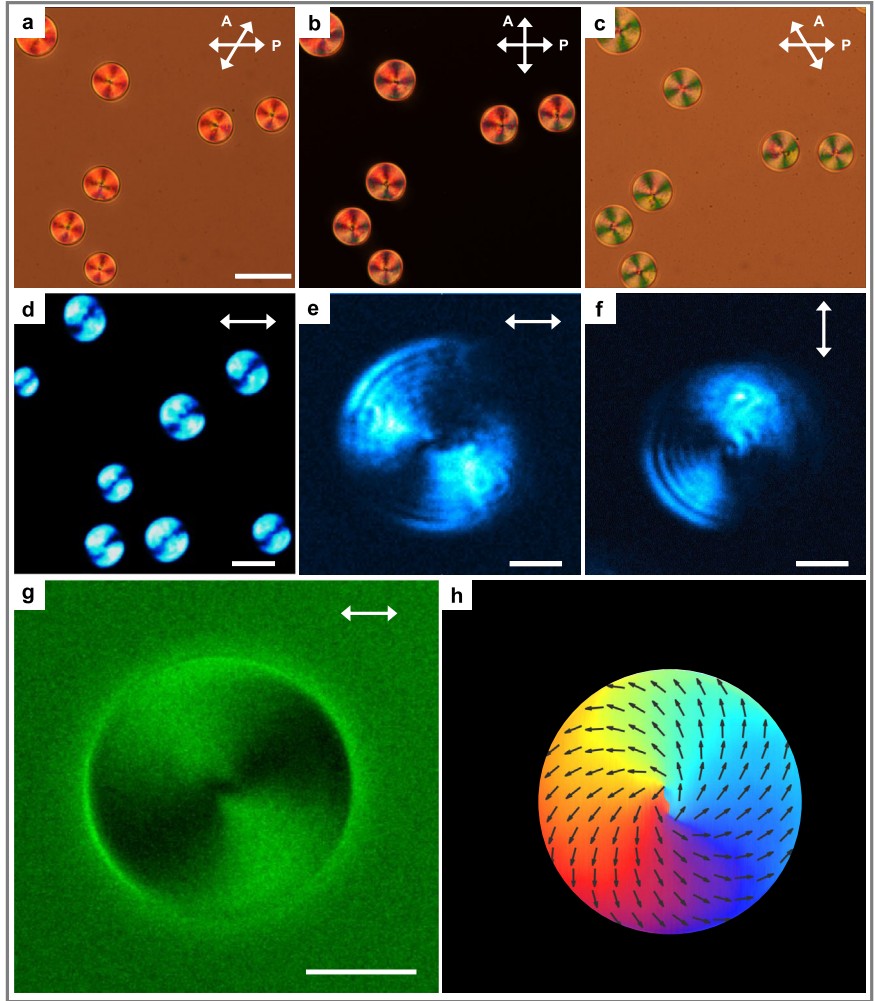

**Fig. 5 | Chirality biasing of $N_F$ droplets. a–c** PLM texture of a RM-OC$_2$/R811 mixture (wt% = 99.5/0.5) taken under different combinations of polarizers. The analyzer is rotated 31°Clockwise (**a**) and 31°anticlockwise (**c**). Scale bar, 50 μm. **d** XY-cross-sectional SHG-CPM image in the midplane of the cell visualized by a linearly polarization **P**. Scale bar, 20 μm. **e, f** SHG microscopy images. Scale bars, 3 μm. **g** XY-cross-sectional FCPM image visualized by a linearly polarization indicated by the white arrow. Scale bar, 10 μm. **h** Reconstructed right-handed divergent polarization fields from SHG-I and FCPM data.

classical apolar nematic droplets[57–60]. Details of the two-dimensional distribution of polarization fields at different sample depth for all the structures is shown in Fig. 6f.

It is important to mention that the experimentally-observed D-meron-like structure appears only when $K_{11}$ satisfies proper conditions: 0.2 pN < $K_{11}$ < 0.8 pN. The variation of $K_{22}$ or $K_{33}$ with respect to other elastic moduli seems to have less potential to induce D-meron (Supplementary Figs. 11–13). Herein, we demonstrate a representative simulated state diagram by varying the effective polarization strength $P_0$ and the elastic anisotropy (Fig. 6g). The elastic anisotropy is defined as a ratio of the splay to bend elasticity $K_{11}/K_{33}$ (assuming the twist and bend elastic modulus are 0.8 pN and 2.0 pN, respectively, as are close to the reported values in the vicinity of the N-$N_F$ phase transition[47]). When the polarity is absent or small in the system (so $P_0 < 2.5 \times 10^{-4}$ C m$^{-2}$), the bipolar structure with a large portion of splay deformation is dominant in small $K_{11}/K_{33}$ values ($K_{11}/K_{33} < 0.4$). With increasing $K_{11}/K_{33}$ (e.g., path I in Fig. 6g), so increasing the penalty of the splay deformation[42,57,58,61], the bipolar structure becomes unstable and is replaced by the escaped vortex-like apolar director field. Further increasing the $K_{11}/K_{33}$ will cause a further transition to an apolar variant of the C-meron. We note that, keeping mind that our system has a curved cylindrical shape like a cylinder LC cell, the corresponding apolar textures in the sphere droplet geometry with tangential anchoring have been well studied in traditional nematic materials by

simulations and experiments[57–59,61–63]. The bipolar structure is easier to form in spherical confined space. It is reported that when $K_{11}/K_{33} \leq 1$, the bipolar structure with two boojums defects can be observed in spherical nematic droplets[62]. Especially, when $K_{11} \geq K_{22} + 0.43K_{33}$, this typical bipolar structure will distort and change to a twisted bipolar structure[58]. If $K_{11}$ is further increased, the most stable energy structure in the spherical droplet is no longer the apolar bipolar structure, but the apolar concentric structure. This tendency in the reports is consistent with the result for $P_0 = 0$ in our cylindrical droplets.

Introducing polarity into the system dramatically changes the director field. In small $K_{11}/K_{33}$ range, when the polarization strength $P_0$ increases (e.g., path II in Fig. 6g), due to the preference of antiparallel packing of local polarizations by the enhanced dipolar interaction, the bipolar or escaped vortex director field changes to the defect-less D-meron. In larger $K_{11}/K_{33}$ range ($K_{11}/K_{33} > 0.4$), the polar version of C-meron with a pure bend (so concentric) replaces D-meron. For C-meron, because the topology is invariant in the depth direction, the topology only carries two possibilities of $[Q, Q_v, Q_h] = [1/2, +1, \pm \pi/2]$, i.e., clockwise or anticlockwise. For both cases, due to the absence of in-plane and out-of-plane twisting structures, they cannot show the non-extinct textures as observed in the experiment (Fig. 3). On the other hand, for D-meron, an in-plane splay-bend deformation and an accompanying spiral streamline of polarization from the center exist. This results in eight species of topology with distinct combinations of

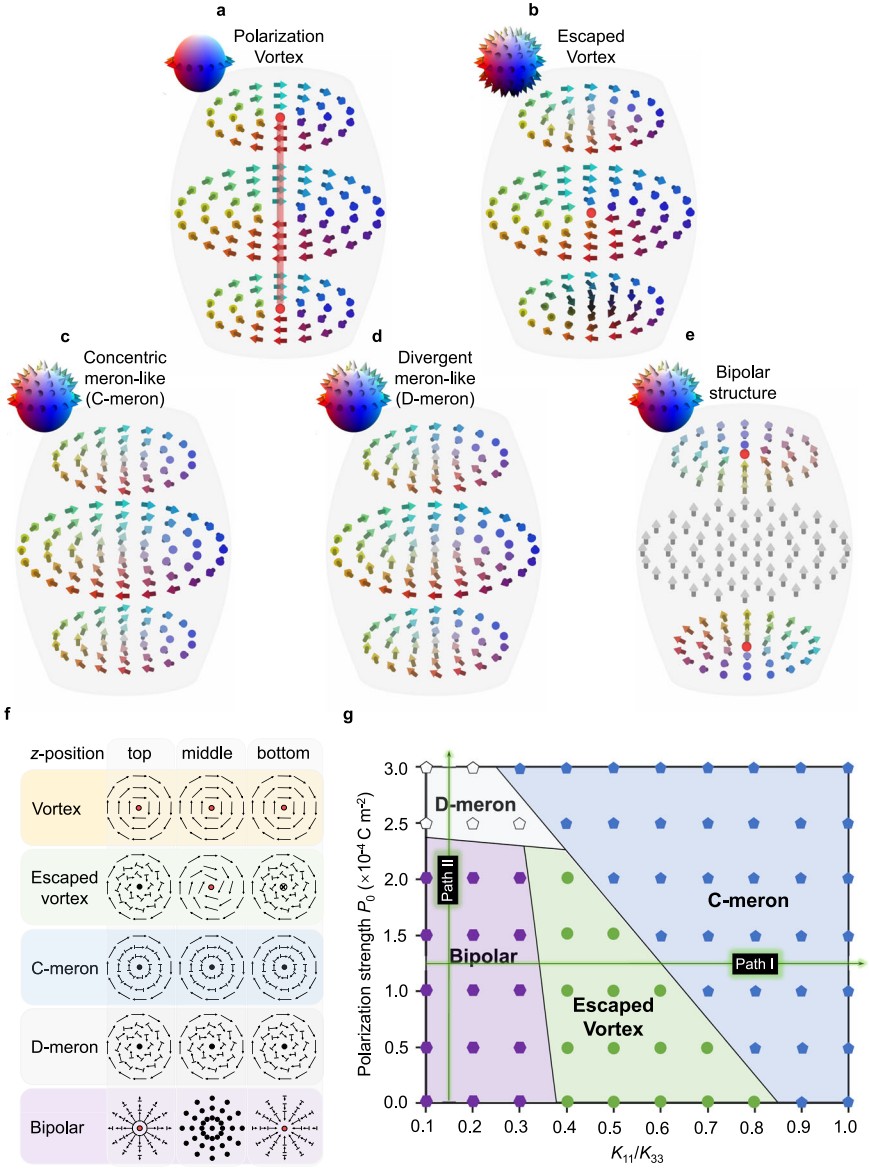

**Fig. 6 | Topological structures of the $N_F$ phase in cylindrically confined space.** **a**–**e** Director fields and the corresponding order parameter space of simple polarization vortex (**a**), escaped vortex (**b**), concentric meron-like structure (**c**), divergent meron-like structure (**d**), and bipolar structure (**e**). **f** The polarization fields projected on the xy-plane in different depth positions of the droplets are described nail vectors. Red points mean point or line defect. **g** State diagram

dependent on the ratio of the splay modulus to the bend modulus ($K_{11}/K_{33}$) and polarization strength $P_0$. The bipolar structure appears in the region where both $K_{11}$ and $P_0$ are small. Otherwise, it is replaced by the escaped vortex or D-meron structure. Regions representing different topological structures are distinguished by different colors.

topological numbers, $[Q, Q_v, Q_h] = [\pm 1/2, +1, \pm \pi/2]$ and clockwise/counter-clockwise spirals. However, the polarization fields of a meron and an anti-meron with opposite handedness are degenerate in experiments since the current nonlinear optical microscopy have no axial resolution for differentiating the polarization states along light propagation direction, i.e., up or down along the viewing direction (Supplementary Figs. 14, 15). Thus, only four types of meron-like structures are experimentally observed (Fig. 4h–k). The appearance necessitates the system possessing a strong polarization strength over the transition threshold, i.e., $P_0 > 2.5 \times 10^{-4}$ C m$^{-2}$.

Summarizing, the real-space observation of polarization meron variants in the fluidic LC state resembles that of magnetic and spin systems. The results suggest the importance of competition between polar and elastic anisotropies. Our findings shed new light on understanding the complex relationship between polar interactions and diverse liquid crystalline orderings, and open the

door for the design and engineering of polarization topologies in liquid matter systems.

## Methods

### Sample preparation

RM-OC$_2$ (Supplementary Fig. 1) is synthesized in the laboratory (Materials and Methods in Supplementary Information). The chirality biasing is made by mixing a chiral dopant, (R)-**2**-Octyl 4-[4-(Hexyloxy) benzoyloxy]benzoate (R811, TCI) or (S)-**2**-Octyl 4-[4-(Hexyloxy)ben-zoyloxy]benzoate (S811, TCI) with RM-OC$_2$. The concentrations of the chiral dopants are adjusted in the range of 0.05–1 wt%.

### LC slab preparation

We introduce samples to commercial or homemade LC slabs with different thicknesses. The glass substrates of LC slabs are spin-coated by a planar alignment layer of KPI-3000 (Shenzhen Haihao Technology

Co., Ltd). No rubbing process is made, so the surface anchoring corresponds to the degenerate planar. For PLM observation, we use homemade planar LC slabs with thicknesses of 3.2 μm, 5 μm, and 10 μm. For the measurement of the SH efficiency and the observations under SHG, SHG-I, and FCPM microscopies, we use homemade 5-μm and 10-μm thick LC cells with planar alignment. For the measurement of the dielectric constant, homemade 10-μm thick LC slabs with Chromium (Cr) and gold (Au) coated layers are used. There is no further surface treatment on the Cr and Au layers.

### Fluorescence confocal polarizing microscopy

FCPM observation is performed using an inverted microscope equipped with a halogen illuminator (Zeiss HAL 100 illuminator). The temperature of samples is controlled by a homemade temperature sink and controller. A ×63 oil-immersion objective lens with numerical aperture NA = 1.4 and a ×40 objective lens with numerical aperture NA = 0.95 are used. We use fluorescent dye 9-(Diethylamino)-5H-benzo[a]phenoxazin-5-one (Nile Red, Sigma-Aldrich) which is doped into the RM-OC$_2$ or RM-OC$_2$R811(S811) mixture. The concentration of the fluorescent dye is 0.05 wt%. The Nile Red molecules align parallel to the LC director. We use a linear polarization for the excitation of the dye. The wavelength of the incident laser is 514 nm. The detected fluorescence intensity depends on the angle α between the polarization direction and the director. Thus, by collecting the fluorescence intensity at four polarization conditions of $\alpha_1 = 0$, $\alpha_2 = \pi/4$, $\alpha_3 = \pi/2$, and $\alpha_4 = 3\pi/4$, we determine the orientation of local directors $\mathbf{n} = (\cos\psi\cos\theta, \sin\psi\cos\theta, \sin\theta)$. $\psi$ is the azimuth angle measured from the $x$ axis in the xy-plane and $\theta$ the out-of-plane polar angle measured from the xy-plane. The detected fluorescence intensity follows the equation:

$$I_k = I_{back} + I_{norm}\cos^4\left(\psi - \frac{k\pi}{4}\right)\cos^4\theta. \qquad (3)$$

$k$ takes a range from 0 to 3 and the angle between polarization directions and $x$ axis is $k\pi/4$. $I_{back}$ is the background intensity and $I_{norm}$ the normalized fluorescence. $I_k$ is the transmittance at the polarization condition of $\alpha_k$. The angle $\psi$ is thereby calculated by $\psi = \frac{1}{2}\tan^{-1}\frac{I_1 - I_3}{I_0 - I_2}$. Then, $\theta$ is determined by substituting the calculated $\psi$ into Eq. (3). However, after substitution, the sign of $\theta$ remains unknown. Assuming the polarization field is continuous as also confirmed by SHG-I microscopy, we can deduce the angle $\theta$ in 3D. The angle $\psi$ lies in the range of $[0,\pi]$, which means FCPM does not differentiate the head or tail of polarization. As a result, FCPM gives only the non-vectorized orientational field. The polarization field is deduced by combining SHG-I data and the 3D non-vectorized orientational field probed by FCPM as explained below.

### SHG and SHG interference (SHG-I) measurement

A Q-switched pulsed laser (MPL-III-1064-20μJ, CNI Laser; central wavelength: 1064 nm; maximum power: 200 mW; pulse duration: 5 ns; repetition rate: 100 Hz) is used as the fundamental light. The fundamental beam is set to be polarized and directed into LC cells. The SH light is detected in the transmission geometry by a photomultiplier tube (DH-PMT-D100V, Daheng Optics) or a scientific CMOS camera (Zyla-4.2P-USB3, Andor). For the temperature scanning, the SH signal is recorded every 1 °C under the control of a homemade Labview program. In the SHG interferometry measurement, a Y-cut quartz plate is used as a reference of the SH signal. The SH light from the quartz plate interferes with the SH light from samples. To change the phase difference between the two SH signals, two fused silica plates are put after the quartz plate. They are rotated in the opposite directions to ensure the optical paths unchanged during their rotation. The larger rotation angle of the fused silica plates corresponds to the larger phase difference. For imaging our topological structures, we use fixed rotation angles of the silica plates to obtain best imaging contrast. We also use a commercial inverted microscope (LSM880, Zeiss) equipped with a femtosecond laser (Integrated pre-compensation 80 MHz, Vision II, Chameleon system) for high-speed large area visualization. The wavelength of the incident laser is 800 nm.

### Dielectric spectroscopy

The dielectric properties of samples are characterized by an LCR meter (4284A, Agilent). A homemade Labview program is used for collecting the frequency and temperature dependencies of the dielectric constant.

### Numerical modeling

As described in the main text, we consider the free energy density contributions from the dipolar interaction ($f_d$) besides the Landau ($f_L$), elastic ($f_e$), surface anchoring ($f_s$), and defect core ($F_{core}$) energies:

$$F = \int(f_L + f_e + f_d)\mathrm{d}V + \int_\Omega f_s \mathrm{d}\Omega + F_{core}, \qquad (4)$$

$$f_L = \frac{a}{2}|\mathbf{P}|^2 + \frac{b}{2}|\mathbf{P}|^4, \qquad (5)$$

$$f_e = \tfrac{1}{2}K_{11}(div\,\mathbf{n})^2 + \tfrac{1}{2}K_{22}(\mathbf{n}\cdot(curl\,\mathbf{n}))^2 + \tfrac{1}{2}K_{33}(\mathbf{n}\times(curl\,\mathbf{n}))^2, \qquad (6)$$

$$f_d = \frac{1}{8\pi\varepsilon_0\varepsilon_r}\int\left\{\frac{\mathbf{P}(\mathbf{r}')\cdot\mathbf{P}(\mathbf{r})}{|\mathbf{r}-\mathbf{r}'|^3} - \frac{3[\mathbf{P}(\mathbf{r}')\cdot(\mathbf{r}-\mathbf{r}')][\mathbf{P}(\mathbf{r})\cdot(\mathbf{r}-\mathbf{r}')]}{|\mathbf{r}-\mathbf{r}'|^5}\right\}\mathrm{d}V, \qquad (7)$$

$$f_s = \frac{1}{2}W_S(\cos\Theta - \mathbf{n}\cdot\mathbf{v})^2. \qquad (8)$$

The Landau energy, $f_L$, is summed up to the fourth order, describing the stability of the N$_F$ state. Namely, the deeper potential-well at a finite polarization $\mathbf{P} = P_0\mathbf{n}$ corresponds to a more stable ferroelectric state. $P_0$ is the polarization strength of polar molecules. The elastic energy term, $f_e$, treats the traditional nematic elastic functionals, where only the terms that are quadratic in $\mathbf{n}$ are used[49,64]. It penalizes the splay, twist, and bend elastic deformations through the elastic constants of $K_{11}$, $K_{22}$, and $K_{33}$. The dipolar interaction, $f_d$, describes the electrostatic contributions from all the interacting polar bodies in the system, which includes both the space charge arising from the splay polarization field, $\nabla\cdot\mathbf{P}(\mathbf{r})$[65], and the boundary charge due to the discontinuity of the polarization on the interface. $\varepsilon_0$ is the vacuum permittivity, $8.85 \times 10^{-12}$ C V$^{-1}$ m$^{-1}$. $\varepsilon_r$ is the relative permittivity, which is set to be $10^4$ in our simulations. The value is comparable to the value reported in the literatures[27,48]. Here, due to a constant of $\varepsilon_r$ is used for calculation, though the contribution of the variation of the polarization field is considered, the orientation-dependent dielectric interaction is not fully accounted in the present form. $W_S$, $\mathbf{v}$, and $\Theta$ in the surface anchoring energy are the anchoring coefficient, the normal vector of the droplet surface, and the deviation angle between the polarization and surface normal vectors. Due to the surface charge and surface tension effects, we assume that the Iso-N$_F$ and N$_F$-glass interfaces pose a degenerate anchoring, where director field lies in the interface plane but without specific azimuthal orientational constraint. To reasonably consider this, we employ the degenerate planar anchoring condition[66], i.e., $\Theta = \pi/2$ and $W_s = 10^{-6}$ J/m$^2$. According to Eq. (4), the free energy terms are integrated using the finite volume and surface area elements $\mathrm{d}V = \mathrm{d}x\mathrm{d}y\mathrm{d}z$ and $\mathrm{d}\Omega = \mathrm{d}x\mathrm{d}y$, respectively. The defect core free energy, $F_{core}$, for three defect-holding polarization states, i.e., simple polarization vortex, escaped polarization vortex and bipolar structure, are calculated:

$F_{core} = \pi KL\ln(R_1/a)$ (for simple polarization vortex)[42],
$F_{core} = 8/3\pi KR_2$ (for escaped polarization vortex)[67],
$F_{core} = 8\pi KR_2$ (for bipolar structure)[67].

$K$ is taken as the average of the splay, twist, and bend elastic moduli. $R_1$ and $R_2$ represent the radius of line defects and point defects, respectively. $L$ is the length of line defects. $a$ is the average molecular size. The polar vortex contains a line defect (Fig. 6a), whose length is equal to the sample thickness. Under this condition, the calculated energy is comparable to the energy of the line defect reported in apolar LC systems[42]. The escaped vortex and bipolar structures carry point defects (Fig. 6b and Fig. 6e). The core radius of the defects is assumed to be comparable to the nematic correlation length, i.e., $R_1 = R_2 = 15\,nm$[68]. In the simulation, we create the 3D curved cylindrical geometries in Creo Parametric 4.0, which is nearly consistent with the FCPM experimental observation (Supplementary Fig. 5). The 3D models are built to have orthogonal meshes and are used to run the n-director simulations. We set the maximum and minimum diameters of the curved cylinder to be 10 and 5 μm, respectively, which locates in the middle plane of droplets and at the interface between the glass and LCs. The sample thickness is set to be 5 μm. We perform the relaxation minimization of the corresponding Ginzburg–Landau functionals based on the finite element difference method (within a 3D space of 14 μm × 14 μm × 17 μm). The 3D space is orthogonally divided by grids with the grid spacing in the range from 10 nm to 500 nm. In this range, we make sure that the numerically-calculated topology is consistent, where the topological details do not depend on the size of the grid spacing. To investigate how the strength of polarity and elasticities would affect the topological nature, we vary $P_0$ and the ratio of elasticities, $K_{11}/K_{33}$, for the simulations. Note that, unlike the simulation by using the Landau–de Gennes Q-tensor[42] where the order parameter can change during the orientational relaxation, the predesignated $P_0$ for each simulation condition in the present free-energy context is unchanged during simulation runs since both the polarity strength (so the order parameter of polarization) and the nematic order parameter are constants, i.e., $|\mathbf{P}| = P_0$ and $|\mathbf{n}| = 1$. Under this circumstance, the Landau energy in Eq. (5) is a constant for a chosen combination of ($P_0$, $K_{11}/K_{33}$). This means that the term will not affect what topological states would be chosen. Therefore, in our simulation, the Landau term is discarded. Considering, in our experiment, the $N_F$ droplets appear from the disordered Iso state, we set the initial condition of the director field to be random for most cases. For the conditions at $P_0 < 1 \times 10^{-4}\,C\,m^{-2}$ and $K_{11}/K_{33} > 0.5$, the simulations end to ideal relaxed topologies as shown in Fig. 6g. However, when the polarization strength is large ($P_0 > \sim 1 \times 10^{-4}\,C\,m^{-2}$) and $K_{11}$ is small ($K_{11}/K_{33} < \sim 0.5$), which corresponds to the left corner of the state diagram, a random initial condition would result in somehow disordered results by some trapped defects. For these situations, we use the relaxed structures of the simulation at slightly smaller $P_0$ or larger $K_{11}$ as initial condition for the numerical relaxations. We relax the systems to the equilibrium state, where the free energy is the global minimum. We note that the initial condition does not change the equilibrium topology.

### Reporting summary
Further information on research design is available in the Nature Portfolio Reporting Summary linked to this article.

## Data availability
All data that support the findings of this study are available in the article and in Supplementary Materials. Additional information is available from the corresponding author upon reasonable request. Source data are provided with this paper.

## Code availability
The codes used for the numerical calculations are available from the corresponding author upon reasonable request.

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

## Acknowledgements

S.A. and M.H. acknowledge the National Key Research and Development Program of China (No. 2022YFA1405000), the Recruitment Program of Guangdong (No. 2016ZT06C322), and the 111 Project (No. B18023). S.A. acknowledges the supports from the International Science and Technology Cooperation Program of Guangdong province (No. 2022A0505050006) and General Program of Guangdong Natural Science Foundation (No. 2022A1515011026). M.H. acknowledges the support from the National Natural Science Foundation of China (NSFC No. 52273292).

## Author contributions

S.A. designed and directed the research. J.Y. made measurements and analyses of structure, optical and dielectric properties. J.L. and M.H. synthesized LC materials. Y.Z. and W.T. calculated the director and polarization field, and simulated the optical properties. M.H. and S.A. wrote the manuscript. All the authors discussed and amended about the manuscript.

## Competing interests

The authors declare no competing interests.
