## [Peer Review File · Nature Communications]

Spontaneous electric-polarization topology in confined ferroelectric nematicsREVIEWER COMMENTS

Reviewer #1 (Remarks to the Author):

In the manuscript »Spontaneous electric-polarization topology in confined ferroelectric nematics«, the authors report on the topological objects that form at the phase transition from isotropic to ferroelectric nematic phase. They deduced the structure of the objects using polarized optical microscopy, fluorescence confocal polarizing microscopy, SHG, and SHG-I microscopy. They experimentally showed that the structure of the 2D droplet is consistent with the topology of a meron. They also model the system by minimization of the total free energy of the system. I find the work interesting, but some major issues need to be corrected. The first is technical, it seems that in the pdf I received, the introduction is missing (the reference numbers start at 4). The second is about the way the electrostatic energy of the system is incorporated into the model. So before I can recommend the paper for publication in Nat Comm these issues must be amended.

1) Eq. (7): The electrostatic (self)energy is the energy due to the electric field caused by the polar body affecting the polar body itself. Its microscopic origin is the dipolar interaction between dipoles, so technically, it can be calculated as the sum of the dipolar interaction energy between all the pairs of the dipoles. I assume that is the origin of the f_d term. If it is, it is incorrect as the dipolar interaction has two terms (besides the one proportional to $p_1 \cdot p_2$, there is also a term proportional to $(p_1 \cdot r)(p_2 \cdot r)$, where r is the vector connecting the centers of the dipoles). In the macroscopic picture, when a polar body is described by polarization vector, then the electric field E of all other dipoles acting on a dipole at a certain position can be calculated using bound charges (which is mathematically the most convenient way to use), and the electrostatic energy is the volume integral of $-E \cdot P/2$. So the electrostatic energy can be calculated either way and the result should be the same. So there are not two different contributions (a dipolar one and the one from the bound charges), they are the same because the bound charges come from the dipoles. Moreover, the electrostatic energy can be calculated without assuming an unknown proportionality constant (eg δ in Eq. (7)).

2) The space charge density is not given as a solution of the Eq. (10), it is given by $-\text{Div}.P$ and from it, the (depolarization) field is calculated using Eq.(10). Eq.(10) is correct if the dielectric tensor is isotropic and independent of the position.

3) Is the shape of a 2D droplet used in the calculations cylinder? Please specify.

4) Topological objects such as skyrmions, hopfions, merons ... are usually discussed as objects in a uniform background vector field. In your case, the background is isotropic, so I am not sure that calling them merons is in this sense correct. Maybe it would be better to describe them as meron-like objects.

5) Do the merons annihilate as was observed for example in the ferromagnetic case (N. Gao, S.-G. Je, M.-Y. Im, J. W. Choi, M. Yang, Q. Li, T. Y. Wang, S. Lee, H.-S. Han, K.-S. Lee, W. Chao, C. Hwang, J. Li, and Z. Q. Qiu, "Creation and annihilation of topological meron pairs in in-plane magnetized films," Nat Commun, 10, 5603 (2019).)

Minor issues:

6) In Eq (1) the Pontryagin number is denoted by N , while later in the text the system is characterized by topological numbers $[Q, Q_v, Q_h]$. I assume Q is equivalent to N .

7) Fig 2 a, at which frequency the dielectric constants shown in Fig 2a was measured?

Reviewer #2 (Remarks to the Author):

See attached PDF file for review.

Reviewer #3 (Remarks to the Author):

This article titled “Spontaneous electric-polarization topology in confined ferroelectric nematics” investigated the chiral polarization merons during the emergence and growth of quasi-2D ferroelectric nematic domains. The authors synthesized new liquid crystalline molecules, which showed the ferroelectric nematic phase (N_f). The spiral texture (meron variants) that appeared during the phase transition of these molecules was investigated with the confocal and second harmonic generation microscopies. In addition, the theoretical approaches well support experimental findings. This study is expected to appeal to a broad readership of Nature Communications. However, some parts need to be supplemented, and the authors should explain the contents of the following questions in the research results before publishing.

- 1) The molecule, RM-OC2, is newly synthesized, and the liquid crystal phase of this molecule should be thoroughly discussed in this article. During the cooling, there might be three phase-transition points (67.1°C, 108.4°C, and 131.4°C). What are their phases in each temperature range? There is a minimal discussion about the phase in this article.
- 2) As a follow-up to question 1, there is an unusual phase transition in Extended Data Fig. 1f-g. After the textures become clear at 69 °C, which might be the isotropic phase, even though the temperature increase, new nucleation occurs. This is a very unusual phenomenon in liquid crystal materials, and the authors should account for it.
- 3) In Figure 2d, the textures are distinct from the conventional nematic, which indicates that the head-to-tail symmetry of the director is spontaneously broken. The author investigated the paper's 3D director and polarization of meron variants. It is necessary to perform 3D analysis (confocal & SHG) and PLM analysis (with de-crossing polarizers) even in the state shown in Figure 2d to discuss more details for the N_f phase.
- 4) In Figure 3l, m, the authors drew the vector field in the cylinder. However, the author discussed that the meron variants deviated from cylindrical shapes. Please check it.
- 5) The vector fields in Extended Data Fig. 9 and Extended Data Fig. 10 are too dense to be seen. Please reduce the number of them and edit them to make them more visible.

By synthesizing a ferroelectric liquid crystal material that exhibits phase transition directly from an isotropic state to a ferroelectric state, the authors demonstrate the formation of polar meron structures in the isotropic background. Their identification of the meron structures is realized by the combination of confocal microscopy and second-harmonic generation microscopy, and the authors show spontaneous chiral symmetry breaking of the meron structures. Their experimental observations are complemented by a Landau-type phenomenological theory incorporating the contribution of polarization into the Frank elasticity.

Polar liquids on their own have been attracting greater interest and the authors' ferroelectric liquid crystal could draw broad attention because of its direct transition from an isotropic state to a ferroelectric state. Formation of topological entities with polar order in liquids could appeal to broad audience and motivate further studies on their manipulation and possible functions.

However, the authors' meron structures are quite similar to those of apolar (but chiral) liquid crystals (Ref. 5) with "head" being consistently assigned to the orientational order. Spontaneous chiral symmetry breaking of similar structures in an achiral and apolar liquid crystal has been reported [J. Jeong et al. PNAS **112**, E1837 (2015); Ohzono et al., Sci. Rep. **7**, 16814 (2017)], and it is not clear whether and how polarity plays a role in chiral symmetry breaking. I do not think the authors have shown, in a convincing manner, results that suggest "the importance of competition between polar and elastic anisotropies" (page 9) and cannot be explained without introducing polarity. Moreover, their theoretical argument, together with their presentation of topological classification, is far from satisfactory as I mention below. Therefore, I do not recommend this work for publication in Nature Communication.

[Numerical Modelling (pp. 12 and 13)]

- The value of the parameters in the free energy densities (eqs. 4-8) are not at all given. In page 8, the authors say $\delta = 10^8$, but what is the unit of δ ? The dimension of β in eq. (7) is obviously different from that of d . Nevertheless, their ratio appears in the vertical axis of Fig. 6d as if it were a dimensionless quantity.
- Eq. (9) is the same as eq. (10) because in the static case $\mathbf{E} = -\nabla\varphi$ (the

minus sign should be deleted in eq. (10)). Still, the authors say they determine φ by eq. (9) (which requires ρ) and that ρ is determined by the same equation, now in the form of (10). This is quite unreasonable. I also note that eqs. (9) and (10) regard the liquid crystal as an isotropic material.

- The grid spacing is $\simeq 0.5 \mu\text{m}$, much larger than the typical size of defect cores of a nematic liquid crystal, $\simeq 10 \text{ nm}$. So I am wondering whether the evaluation of the free energy of the liquid crystal involving a line disclination (“Vortex”) has indeed been appropriately carried out; large grid spacings could underestimate the contribution of defect cores, and then Fig. 6b could be significantly affected. I note that an ingenious way of circumventing such difficulties is presented in Stark, Eur. Phys. J. B **10**, 311 (1999). The relaxation from a random initial condition should generate many defects initially, and I am wondering whether the system really relaxes to one of the meron structures without such defects being pinned at numerical grid points.
- In the free energy densities involving \mathbf{P} , do the authors simply replace \mathbf{P} by $P_0\mathbf{n}$ and minimize the free energy functional in terms of \mathbf{n} alone? Then, how is P_0 determined?
- How are curved boundaries between the isotropic region and the ferroelectric region implemented in the numerical calculations, and what are the boundary conditions there?

[Characterization of topological structures]

- In eq. (1), the authors should explicitly state that \mathbf{n} is a unit vector, and what x and y are.
- Is Q that first appears three lines after eq. (2) the same as N defined by eq. (1)?
- Is φ in the second line of eq. (2) different from ϕ in the first line? If it is, what is φ ? If not, eq. (2) looks very strange.
- When the authors first introduce the set $[Q, Q_v, Q_h]$, visualization of examples of structures corresponding to specific sets of $[Q, Q_v, Q_h]$ will help interested readers understand the meaning of these quantities.

- After the authors start discussions on their experimental and numerical results, characterization by $[Q, Q_v, Q_h]$ is made only at the 1st line of page 9. Why do the authors not characterize other structures in the figures by $[Q, Q_v, Q_h]$?
- In page 9, the authors mention that $Q = 1$ in the “escaped-vortex” structure shown in Fig. 6c. If Q is in fact the same as N defined by eq. (1) (see my previous comment), the integration $\int dx dy$ in one plane would yield $Q = \pm 1/2$. What characterization would yield $Q = 1$?
- I believe that the gray spheres in Fig. 6 and Extended Data Figures 9 and 10 represent the order parameter space S^2 . In this representation of S^2 , the value of the order parameter is given by the location of a point on the sphere surface (to be more precise, the order parameter is given by a vector pointing to a point on the sphere from the sphere center). Then I am wondering what the distribution of vectors on the sphere surface means. The authors do not seem to understand the distinction between the distribution of the order parameter in the real space and that in the order parameter space (See, e.g., Figure 12.20 of Kleman and Lavrentovich, “Soft Matter Physics: An introduction”).
- The “achiral meron” illustrated in Fig. 6d is obviously chiral. In the authors’ “chiral merons” there, splay-bend distortions are involved in the radial directions while they are not in “achiral merons”. This seems to contradict the authors’ explanation in page 8, “This is because filling the space with more ratios of twisting structures costs less elastic energy.” A more convincing argument could be given by close inspection of the free energy densities (eqs. 4-8) and the electrostatic potential φ .

[Other issues]

- Figures do not appear in numerical order (For example, in page 6, Figs. 6ab are referred to before Figs 4 and 5).
- The first four lines of page 2 exactly match part of the Abstract.
- While in eqs. (4-8) the director \mathbf{n} and the polarization \mathbf{P} are treated independently, in page 3 the authors say that the director itself is polar ($\mathbf{n} \neq -\mathbf{n}$). If so, the Frank free energy (6) might need modification to incorporate terms odd in \mathbf{n} . The authors should therefore clarify

whether the director is treated as polar, or it is still apolar and the polarity is introduced as \mathbf{P} .

- Is the “space polarization charge” introduced in page 4 equivalent to N (or Q)?
- [Page 5] The word “extinct” is an adjective, not a verb.
- Are SHG-CPM and SHG-I developed by the authors for the first time for this work? If not, references on them should be cited.
- Should I_n be I_k in eq. (3)?
- The phrase “short-range antiferroelectric-like ordering” in page 12 might confuse the readers because the dipolar interaction f_d itself is long-ranged.
- Visibility of the chemical formula and “ $\mu = 13.5D$ ” in Fig. 1b is quite poor.
- The arrows in Extended Data Figures 9 and 10 are hard to distinguish visually.

Point-to-Point Reply to the Reviewers

In the following, the statement of the reviewer and our answers/discussions appear in *black* and *blue* fonts, respectively.

To Reviewer 1:

Thank you very much for reviewing our manuscript and we highly appreciate your invaluable and helpful feedbacks to our manuscript. Based on the Reviewer's comments, we reconsidered the free energy density equation of dipole-dipole interactions and the space charge effect and provide new numerical simulation results. We also provide more detail information on dielectric measurement and fluorescence confocal polarizing microscopy results. We answered your questions and significantly revised our manuscript as stated in the following according to your comments and suggestions. A comprehensive revision list can be found at the end of the reply. We hope that you are satisfied with our replies and revision. Your positive decision would be very much appreciated.

1) Eq. (7): The electrostatic (self)energy is the energy due to the electric field caused by the polar body affecting the polar body itself. Its microscopic origin is the dipolar interaction between dipoles, so technically, it can be calculated as the sum of the dipolar interaction energy between all the pairs of the dipoles. I assume that is the origin of the f_d term. If it is, it is incorrect as the dipolar interaction has two terms (besides the one proportional to $p_1 \cdot p_2$, there is also a term proportional to $(p_1 \cdot r)(p_2 \cdot r)$, where r is the vector connecting the centers of the dipoles). In the macroscopic picture, when a polar body is described by polarization vector, then the electric field E of all other dipoles acting on a dipole at a certain position can be calculated using bound charges (which is mathematically the most convenient way to use), and the electrostatic energy is the volume integral of $-E \cdot P/2$. So the electrostatic energy can be calculated either way and the result should be the same. So there are not two different contributions (a dipolar one and the one from the bound charges), they are the same because the bound charges come from the dipoles. Moreover, the electrostatic energy can be calculated without assuming an unknown proportionality constant (eg δ in Eq. (7)).

-> Thank you for your comments. In the original version, we considered the model should have a similar form with that in the magnetic systems, so that the analogy can be discussed (Kudo, K., *et al.*, *J. Phys. Soc. Japan* 2007, **76**, 013002; Kudo, K., *et al.*, *Phys. Rev. E* 2009, **80**, 016209; Anzaki, R., *et al.*, *Phys. Rev. B* 2021, **103**, 094408.). However, after reading the comments from you and Reviewer 2, we felt we must reconstruct the free energy primarily based on dipolar interaction term. In addition, though the simple dipolar term also includes the bound charge effect (surface symmetry breaking) as the reviewer suggested, the polarization-deformation-induced space charge in the bulk is not considered. It is because the resulting electric field is not acting on the dipole itself, but the space charge caused by the divergence of a polarization field. Therefore, in the revised version, we

considered the dipolar potential based on: $f_{DDI} = \frac{1}{4\pi\epsilon_0\epsilon_r} \int \frac{\mathbf{P}(\mathbf{r}') \cdot \mathbf{P}(\mathbf{r})}{|\mathbf{r} - \mathbf{r}'|^3} - \frac{3[\mathbf{P}(\mathbf{r}') \cdot (\mathbf{r} - \mathbf{r}')][\mathbf{P}(\mathbf{r}) \cdot (\mathbf{r} - \mathbf{r}')]}{|\mathbf{r} - \mathbf{r}'|^5} dV$

as well as the space charge $f_{SC} = \frac{1}{4\pi\epsilon_0\epsilon_r} \int \frac{\nabla \cdot \mathbf{P}(\mathbf{r}') \cdot \nabla \cdot \mathbf{P}(\mathbf{r})}{|\mathbf{r} - \mathbf{r}'|} dV$. Moreover, according to the Reviewer's

comment, we discarded the unknown proportionality constant in new numerical calculations, only the “realistic” polarization strength P_0 and elastic modulus K_i as variables to investigate the competition between polar and elastic anisotropies. We also apologize that, in the original free energy description of the elastic deformation, the chiral term was wrongly added in the manuscript (the current systems are achiral as described). Now, it is corrected. We redraw state diagrams based on polarization strength and elastic anisotropies and made a comprehensive argument based on the results. Therefore, we significantly rewrote the discussion sections and numerical modelling in the method sections. The comprehensive revisions can be found in the highlighted manuscript.

Page 14, Line 6

(Before) “**Numerical Modelling.** As described in the main text, we consider the free energies composed of the polar gradient (f_p), dipolar interaction (f_d) and the space charge (f_{sc}) contributions besides the Landau (f_L), elastic (f_e) and surface anchoring (f_s) energies:

$$F = \int (f_L + f_e + f_f) dV + \int_{\Omega} f_s d\Omega, \quad (4)$$

$$f_L = \frac{a}{2} |\mathbf{P}|^2 + \frac{b}{2} |\mathbf{P}|^4, \quad (5)$$

$$f_e = \frac{1}{2} K_{11} (\text{div } \mathbf{n})^2 + \frac{1}{2} K_{22} (\mathbf{n} \cdot (\text{curl } \mathbf{n}))^2 + \frac{1}{2} K_{33} (\mathbf{n} \times (\text{curl } \mathbf{n}))^2 - \frac{1}{2} K'_{22} \mathbf{n} \cdot (\nabla \times \mathbf{n}), \quad (6)$$

$$f_f = f_p + f_d + f_{sc} = \frac{d}{2} (\nabla \mathbf{P})^2 + \beta \int \frac{\mathbf{P}(\mathbf{r}) \cdot \mathbf{P}(\mathbf{r}')}{|\mathbf{r} - \mathbf{r}'|^3} d^3r + \delta(\partial_i \varphi) \mathbf{P}_i, \quad (7)$$

$$f_s = \frac{1}{2} W_S (\cos \Theta - \mathbf{n} \cdot \mathbf{v})^2. \quad (8)''$$

(After) “**Numerical Modelling.** As described in the main text, we consider the free energy density contributions from the dipolar interaction (f_d) and the space charge (f_{sc}) contributions besides the Landau (f_L), elastic (f_e), surface anchoring (f_s) and defect core (F_{core}) energies:

$$F = \int (f_L + f_e + f_f) dV + \int_{\Omega} f_s d\Omega + F_{\text{core}}, \quad (4)$$

$$f_L = \frac{a}{2} |\mathbf{P}|^2 + \frac{b}{2} |\mathbf{P}|^4, \quad (5)$$

$$f_e = \frac{1}{2} K_{11} (\text{div } \mathbf{n})^2 + \frac{1}{2} K_{22} (\mathbf{n} \cdot (\text{curl } \mathbf{n}))^2 + \frac{1}{2} K_{33} (\mathbf{n} \times (\text{curl } \mathbf{n}))^2, \quad (6)$$

$$f_d = \frac{1}{4\pi\epsilon_0\epsilon_r} \int \frac{\mathbf{P}(\mathbf{r}') \cdot \mathbf{P}(\mathbf{r})}{|\mathbf{r} - \mathbf{r}'|^3} - \frac{3[\mathbf{P}(\mathbf{r}') \cdot (\mathbf{r} - \mathbf{r}')][\mathbf{P}(\mathbf{r}) \cdot (\mathbf{r} - \mathbf{r}')]}{|\mathbf{r} - \mathbf{r}'|^5} dV, \quad (7)$$

$$f_{sc} = \frac{1}{4\pi\epsilon_0\epsilon_r} \int \frac{\nabla \cdot \mathbf{P}(\mathbf{r}') \cdot \nabla \cdot \mathbf{P}(\mathbf{r})}{|\mathbf{r} - \mathbf{r}'|} dV, \quad (8)$$

$$f_f = f_d + f_{sc}, \quad (9)$$

$$f_s = \frac{1}{2} W_S (\cos \theta - \mathbf{n} \cdot \mathbf{v})^2. \quad (10)''$$

Page 14, Line 18

(Sentence added): “ P_0 is the intrinsic polarization strength of polar molecules.”

2) The space charge density is not given as a solution of the Eq. (10), it is given by $-\text{Div.P}$ and from it, the (depolarization) field is calculated using Eq.(10). Eq.(10) is correct if the dielectric tensor is isotropic and independent of the position.

-> We thank the reviewer for pointing out this issue. We also apologize for the insufficient description for this part. The space charge density is related to the divergence of polarization fields, $-\nabla \cdot \mathbf{P}(\mathbf{r}) = \rho(\mathbf{r})$. The Maxwell equation of the original equation 10 in the original manuscript is incomplete to describe what we have made in the calculation. Indeed, the electric field \mathbf{E} is generated by the polar body so the electric displacement can only be interpreted by the polarity \mathbf{P} in the system. Therefore, we can calculate the space charge density by, $-\nabla \cdot \mathbf{P}(\mathbf{r}) = \rho(\mathbf{r})$, and correct our space charge term to be $f_{sc} = \frac{1}{4\pi\epsilon_0\epsilon_r} \int \frac{\nabla \cdot \mathbf{P}(\mathbf{r}') \cdot \nabla \cdot \mathbf{P}(\mathbf{r})}{|\mathbf{r}-\mathbf{r}'|} dV$. Also, as the reviewer suggests, the current formulism, though considers polarization fields correctly, is of an isotropic because the dielectric permittivity is a constant. As in our discussion, we rewrote all the part and discussed the potential role of the space charge, and the addition of the term would not significantly change state diagram as calculated. The exploration of the full description of the anisotropic electrostatic interactions in polar liquid crystals is beyond the current scope of the experimental discovery of polarization topology and should be further discussed in the community in future. Therefore, we revised the manuscript to explain the circumstances. As below, we show some related major revision in the manuscript. The comprehensive revisions can be found in the highlighted manuscript.

Page 8, Line 18, Discussion section

(Sentence Added) Please refer to the rewritten Discussion part in the revised manuscript.

Page 11, Line 13

(Sentence added) “We note that, though the current calculation of the space charge term considers the contribution from the variation of the polarization field that dislikes the splay deformation, the isotropic distribution of the dielectric permittivity in space is considered through using a dielectric constant of ϵ_r . The exploration of the full description of the anisotropic electrostatic interactions in polar liquid crystals is beyond the current scope and will be explored in future. However, considering the role of space charge revealed above, we expect the full introduction of the anisotropic dielectric interaction may only work to slightly shift the phase boundaries.”

Page 14, Line 20

(Sentence added) “The free energy term, f_f , is the total polar terms including the dipolar interaction, f_d , and space charge effect, f_{sc} . The dipolar interaction, f_d , describes the electrostatic contributions from all the interacting polar bodies in the system, which includes the boundary charge due to the

discontinuity of the polarization on the interface. When a polarization field in the bulk is inhomogeneous, it creates ‘virtual’ charges (called space charge). Especially, the splay polarization field $\nabla \cdot \mathbf{P}(\mathbf{r})$ is the most energetically unfavorable. To include the space charge effect, f_{sc} , we calculate the electrostatic contributions from all the interacting space charges in the bulk⁶². ϵ_0 is the vacuum permittivity, $8.85 \times 10^{-12} \text{ C V}^{-1} \text{ m}^{-1}$, ϵ_r is the dielectric constant of materials. Here, due to a constant of ϵ_r is used for calculation, though the contribution of the variation of the polarization field is considered, the orientation-dependent dielectric interaction is not fully accounted in the present form.”

3) Is the shape of a 2D droplet used in the calculations cylinder? Please specify.

-> We apologize for the confusion caused by Figures 3l-m. The shape of the droplet used in calculations is 3D curved cylindrical shape as revealed by confocal microscopy. The size of the used cylinder is equal to the geometry and the size of droplet observed in FCPM (Extended Figure 3). The XY-cross section of the cylinder on the interface between glass and N_F LC is a circular shape with radius of 5 μm . The XY-cross section in the middle plane is a circular shape with radius of 10 μm . The sample thickness is 10 μm . To state more clearly, we revised the droplet shape in Figures 3l-m and described the used shape geometry for calculations more detailed.

Page 15, Line 11

(Before) “...In the simulation, we create the droplet geometries in Creo Parametric 4.0 and import...Considering the size of droplets observed in our experiment, we set the diameter of the model to be 20 μm and the height to be 5 μm . We perform the relaxation minimization of the corresponding Ginzburg–Landau functionals based on the finite element difference method (within a 3D space of 25 $\mu\text{m} \times 25 \mu\text{m} \times 20 \mu\text{m}$, represented by a 60 \times 60 \times 40 grid space). ...”

(After) “...In the simulation, we create the 3D curved cylindrical geometries in Creo Parametric 4.0, which is nearly consistent with the FCPM experimental observation (Extended Figure 3). The 3D models are built to have orthogonal meshes and are used to run the n-director simulations. We set the maximum and minimum diameters of the curved cylinder to be 10 and 5 μm , respectively, which locates in the middle plane of droplets and at the interface between the glass and LCs. The sample thickness is set to be 5 μm . We perform the relaxation minimization of the corresponding Ginzburg–Landau functionals based on the finite element difference method (within a 3D space of 14 $\mu\text{m} \times 14 \mu\text{m} \times 7 \mu\text{m}$). ...”

Page 20, Figures 3l,m

(Before)

(After)

4) Topological objects such as skyrmions, hopfions, merons ... are usually discussed as objects in a uniform background vector field. In your case, the background is isotropic, so I am not sure that calling them merons is in this sense correct. Maybe it would be better to describe them as meron-like objects.

-> We thank the reviewer for pointing this issue. Since merons are topological solitons embedded in a uniform vector field, the topological structures with isotropic background cannot be directly classified as merons. As suggested, we revised our manuscript and changed the name of the object of meron to be meron-like structures.

5) Do the merons annihilate as was observed for example in the ferromagnetic case (N. Gao, S.-G. Je, M.-Y. Im, J. W. Choi, M. Yang, Q. Li, T. Y. Wang, S. Lee, H.-S. Han, K.-S. Lee, W. Chao, C. Hwang, J. Li, and Z. Q. Qiu, "Creation and annihilation of topological meron pairs in in-plane magnetized films," Nat Commun, 10, 5603 (2019).)

-> Thank you for the comments on the annihilation event. In the present ferroelectric nematic case, we have never observed the annihilation. In the LC droplet, the droplet geometry as well as the surface anchoring condition provide a confinement, which is expected to retain the topology. When the merons collide, they tend to form line disclination on the interface to separate the initial meron-like textures (please refer to the figure below).

Confidential Fig. 1. The representative PLM texture of collision of merons. **a**, The PLM texture evolution of RM-OC₂ under crossed polarizers in a homemade LC cell without rubbing. Cell thickness: 5 μm. Scale bar: 100 μm. **b**, The PLM texture evolution of collision of meron-like bodies under crossed polarizers in a homemade LC cell without rubbing. Cell thickness: 10 μm. Scale bar: 30 μm.

6) In Eq (1) the Pontryagin number is denoted by N , while later in the text the system is characterized by topological numbers [Q , Q_v , Q_h]. I assume Q is equivalent to N .

-> We apologize for the confusion made here. As you suggested, the topological numbers Q is equivalent to N , so we replaced N by Q in equation (1) to avoid confusing the readers. Also, we notice we have forgotten to cite an important literature related to LC topology, O. Lavrentovich, Liq. Cryst. 24, 117 (1998). We added the reference as Ref. [21].

Page 2, Line 10

(Before) “The Pontryagin number, N , defines how many times a unit vector that defines the average orientation, the so-called director \mathbf{n} , wraps the order parameter space S^2 (i.e. two-dimensional sphere)^{4, 20}, represented by

$$N = \frac{1}{4\pi} \int \mathbf{n} \cdot (\partial_x \mathbf{n} \times \partial_y \mathbf{n}) dx dy. \quad (1)''$$

(After) “The Pontryagin number, Q , defines how many times a unit vector that defines the average orientation, the so-called director \mathbf{n} , wraps the unit sphere^{4, 20, 21}, represented by

$$Q = \frac{1}{4\pi} \int \mathbf{n} \cdot (\partial_x \mathbf{n} \times \partial_y \mathbf{n}) dx dy. \quad (1)''$$

7) Fig 2a, at which frequency the dielectric constants shown in Fig 2a was measured?

-> We apologize for the insufficient description for the figure. In Fig. 2a, The temperature dependencies of the dielectric constant at a frequency of 25 Hz. The dielectric constant is measurement dependent on the temperature and frequency for samples RM-OC₂.

Page 19, Figure 2

(Before) "...Bottom chart shows the temperature dependencies of the dielectric constant (blue squares). ..."

(After) "... Bottom chart shows the temperature dependencies of the dielectric constant at a frequency of 25 Hz (blue squares). ..."

To Reviewer 2:

Thank you very much for reviewing our manuscript and giving us your comments. We highly appreciate your invaluable and helpful feedbacks to our manuscript. Based on the Reviewer's comments, we reconsidered the free energy density equation of dipole-dipole interactions and the space charge effect and provide new numerical simulation results. We believe that the simulation results show the role of polarity in determining the topology in polar nematic liquid crystals. We also revised the descriptions about theoretical arguments and topology classification. We answered your questions and significantly revised our manuscript as stated in the following according to your comments and suggestions. A comprehensive revision list can be found at the end of the reply. We hope that you are satisfied with our replies and revision. Your positive decision would be very much appreciated.

[Numerical Modelling (pp. 12 and 13)]

1. The value of the parameters in the free energy densities (eqs. 4-8) are not at all given. In page 8, the authors say $\delta = 10^8$, but what is the unit of δ ? The dimension of β in eq. (7) is obviously different from that of d . Nevertheless, their ratio appears in the vertical axis of Fig 6d as if it were a dimensionless quantity.
2. Eq. (9) is the same as eq. (10) because in the static case $\mathbf{E} = -\nabla\varphi$ (the minus sign should be deleted in eq. (10)). Still, the authors say they determine δ by eq. (9) (which requires ρ and that ρ is determined by the same equation, now in the form of (10)). This is quite unreasonable. I also note that eqs. (9) and (10) regard the liquid crystal as an isotropic material.

-> Thank you for these two comments. In the original version, In the original manuscript, δ is dimensionless while the dimension of $\beta = \frac{\beta_0}{4\pi\epsilon_0\epsilon_r}$ is [N m² C⁻²] and the dimension of d is [m⁴ C⁻²], so β/d is not dimensionless. We considered the free energy model should have a similar form with that in the magnetic systems, so that the analogy can be discussed (Kudo, K., *et al.*, *J. Phys. Soc. Japan* 2007, **76**, 013002; Kudo, K., *et al.*, *Phys. Rev. E* 2009, **80**, 016209; Anzaki, R., *et al.*, *Phys. Rev. B* 2021, **103**, 094408.).

However, after reading the comments from you and Reviewer 1, we felt we must reconstruct the free energy primarily based on dipolar interaction term. In addition, though the simple dipolar term also includes the bound charge effect (surface symmetry breaking) as the reviewer suggested, the polarization-deformation-induced space charge in the bulk is not considered. It is because the resulting electric field is not acting on the dipole itself, but the space charge caused by the divergence of a polarization field. Therefore, in the revised version, we considered the dipolar

potential based on: $f_{DDI} = \frac{1}{4\pi\epsilon_0\epsilon_r} \int \frac{\mathbf{P}(\mathbf{r}') \cdot \mathbf{P}(\mathbf{r})}{|\mathbf{r}-\mathbf{r}'|^3} - \frac{3[\mathbf{P}(\mathbf{r}') \cdot (\mathbf{r}-\mathbf{r}')] [\mathbf{P}(\mathbf{r}) \cdot (\mathbf{r}-\mathbf{r}')] }{|\mathbf{r}-\mathbf{r}'|^5} dV$ as well as the space

charge $f_{SC} = \frac{1}{4\pi\epsilon_0\epsilon_r} \int \frac{\nabla \cdot \mathbf{P}(\mathbf{r}') \cdot \nabla \cdot \mathbf{P}(\mathbf{r})}{|\mathbf{r}-\mathbf{r}'|} dV$. Moreover, according to the Reviewers' comments, we

discarded the unknown proportionality constant in new numerical calculations, only the "realistic" polarization strength P_0 and elastic modulus K_i as variables to investigate the competition

between polar and elastic anisotropies. We also apologize that, in the original free energy description of the elastic deformation, the chiral term was wrongly added in the manuscript (the current systems are achiral as described). Now, it is corrected. We redraw state diagrams based on polarization strength and elastic anisotropies and made a comprehensive argument based on the results. Therefore, we significantly rewrote the discussion sections and numerical modelling in the method sections. The comprehensive revisions can be found in the highlighted manuscript.

Page 14, Line 6

(Before) “**Numerical Modelling.** As described in the main text, we consider the free energies composed of the polar gradient (f_p), dipolar interaction (f_d) and the space charge (f_{sc}) contributions besides the Landau (f_L), elastic (f_e) and surface anchoring (f_s) energies:

$$F = \int (f_L + f_e + f_f) dV + \int_{\Omega} f_s d\Omega, \quad (4)$$

$$f_L = \frac{a}{2} |\mathbf{P}|^2 + \frac{b}{2} |\mathbf{P}|^4, \quad (5)$$

$$f_e = \frac{1}{2} K_{11} (\text{div } \mathbf{n})^2 + \frac{1}{2} K_{22} (\mathbf{n} \cdot (\text{curl } \mathbf{n}))^2 + \frac{1}{2} K_{33} (\mathbf{n} \times (\text{curl } \mathbf{n}))^2 - \frac{1}{2} K'_{22} \mathbf{n} \cdot (\nabla \times \mathbf{n}), \quad (6)$$

$$f_f = f_p + f_d + f_{sc} = \frac{d}{2} (\nabla \mathbf{P})^2 + \beta \int \frac{\mathbf{P}(\mathbf{r}) \cdot \mathbf{P}(\mathbf{r}')}{|\mathbf{r} - \mathbf{r}'|^3} d^3r + \delta(\partial_i \varphi) \mathbf{P}_i, \quad (7)$$

$$f_s = \frac{1}{2} W_S (\cos \Theta - \mathbf{n} \cdot \mathbf{v})^2. \quad (8)''$$

(After) “**Numerical Modelling.** As described in the main text, we consider the free energy density contributions from the dipolar interaction (f_d) and the space charge (f_{sc}) contributions besides the Landau (f_L), elastic (f_e), surface anchoring (f_s) and defect core (F_{core}) energies:

$$F = \int (f_L + f_e + f_f) dV + \int_{\Omega} f_s d\Omega + F_{\text{core}}, \quad (4)$$

$$f_L = \frac{a}{2} |\mathbf{P}|^2 + \frac{b}{2} |\mathbf{P}|^4, \quad (5)$$

$$f_e = \frac{1}{2} K_{11} (\text{div } \mathbf{n})^2 + \frac{1}{2} K_{22} (\mathbf{n} \cdot (\text{curl } \mathbf{n}))^2 + \frac{1}{2} K_{33} (\mathbf{n} \times (\text{curl } \mathbf{n}))^2, \quad (6)$$

$$f_d = \frac{1}{4\pi\epsilon_0\epsilon_r} \int \frac{\mathbf{P}(\mathbf{r}') \cdot \mathbf{P}(\mathbf{r})}{|\mathbf{r} - \mathbf{r}'|^3} - \frac{3[\mathbf{P}(\mathbf{r}') \cdot (\mathbf{r} - \mathbf{r}')][\mathbf{P}(\mathbf{r}) \cdot (\mathbf{r} - \mathbf{r}')]}{|\mathbf{r} - \mathbf{r}'|^5} dV, \quad (7)$$

$$f_{sc} = \frac{1}{4\pi\epsilon_0\epsilon_r} \int \frac{\nabla \cdot \mathbf{P}(\mathbf{r}') \cdot \nabla \cdot \mathbf{P}(\mathbf{r})}{|\mathbf{r} - \mathbf{r}'|} dV, \quad (8)$$

$$f_f = f_d + f_{sc}, \quad (9)$$

$$f_s = \frac{1}{2} W_S (\cos \Theta - \mathbf{n} \cdot \mathbf{v})^2. \quad (10)''$$

Page 14, Line 18

(Sentence added): “ P_0 is the intrinsic polarization strength of polar molecules.”

For the treatment of space charge term

We apologize for the insufficient description about this part. In our previous calculation, the energy $f_{sc} = \delta(\partial_i \varphi) \mathbf{P}_i$ is due to the electric field ($\mathbf{E} = -\nabla\varphi$) caused by the polar body \mathbf{P} affecting the polar body itself. We first use equation (9) to determine the electrostatic potential φ which can be used to calculate the electric field \mathbf{E} . In equation (9), the space charge ρ density is determined by the divergence of polarity. Moreover, the proportional constant δ is used to vary the strength of space charge effect.

In our revision, to control the strength of space charge effect, instead of varying the proportional constant δ , we naturally vary the intrinsic polarity P_0 of the material and elastic moduli. Again, the space charge density is related to the divergence of polarization fields. The Maxwell equation of the original equation 10 in the original manuscript is incomplete to describe what we have made in the calculation. Indeed, the electric field \mathbf{E} is generated by the polar body so the electric displacement can only be interpreted by the polarity \mathbf{P} in the system. Therefore, we can calculate the space charge density by, $-\nabla \cdot \mathbf{P}(\mathbf{r}) = \rho(\mathbf{r})$, and correct our space charge term to be $f_{sc} = \frac{1}{4\pi\epsilon_0\epsilon_r} \int \frac{\nabla \cdot \mathbf{P}(\mathbf{r}') \cdot \nabla \cdot \mathbf{P}(\mathbf{r})}{|\mathbf{r} - \mathbf{r}'|} dV$. As the reviewer suggests, the current formulism, though considers polarization fields correctly, is of an isotropic because the dielectric permittivity is constant. As in our discussion, we discussed the potential role of the space charge, and the addition of the term would not significantly change state diagram as calculated. The exploration of the full description of the anisotropic electrostatic interactions in polar liquid crystals is beyond the current scope of the experimental discovery of polarization topology and should be further discussed in the community in future. Therefore, we revised the manuscript to explain the circumstances. As below, we show some related major revision in the manuscript. The comprehensive revisions can be found in the highlighted manuscript.

Page 8, Line 18, Discussion section

(Sentence Added) Please refer to the rewritten Discussion part in the revised manuscript.

Page 11, Line 13

(Sentence added) “We note that, though the current calculation of the space charge term considers the contribution from the variation of the polarization field that dislikes the splay deformation, the isotropic distribution of the dielectric permittivity in space is considered through using a dielectric constant of ϵ_r . The exploration of the full description of the anisotropic electrostatic interactions in polar liquid crystals is beyond the current scope and will be explored in future. However, considering the role of space charge revealed above, we expect the full introduction of the anisotropic dielectric interaction may only work to slightly shift the phase boundaries.”

Page 14, Line 20

(Sentence added) “The free energy term, f_f , is the total polar terms including the dipolar interaction, f_d , and space charge effect, f_{sc} . The dipolar interaction, f_d , describes the electrostatic contributions from all the interacting polar bodies in the system, which includes the boundary charge due to the discontinuity of the polarization on the interface. When a polarization field in the bulk is inhomogeneous, it creates ‘virtual’ charges (called space charge). Especially, the splay polarization field $\nabla \cdot \mathbf{P}(\mathbf{r})$ is the most energetically unfavorable. To include the space charge effect, f_{sc} , we calculate the electrostatic contributions from all the interacting space charges in the bulk⁶². ϵ_0 is the vacuum permittivity, $8.85 \times 10^{-12} \text{ C V}^{-1} \text{ m}^{-1}$, ϵ_r is the dielectric constant of materials. Here, due to a constant of ϵ_r is used for calculation, though the contribution of the variation of the polarization field is considered, the orientation-dependent dielectric interaction is not fully accounted in the present form.”

3. The grid spacing is $\approx 0.5 \mu\text{m}$, much larger than the typical size of defect cores of a nematic liquid crystal, $\approx 10 \text{ nm}$. So I am wondering whether the evaluation of the free energy of the liquid crystal involving a line disclination (“Vortex”) has indeed been appropriately carried out; large grid spacings could underestimate the contribution of defect cores, and then Fig. 6b could be significantly affected. I note that an ingenious way of circumventing such difficulties is presented in Stark, Eur. Phys. J. B 10, 311 (1999). The relaxation from a random initial condition should generate many defects initially, and I am wondering whether the system really relaxes to one of the meron structures without such defects being pinned at numerical grid points.

-> We appreciate your comments on the size of defects and grid size in the numerical calculations. To answer these questions, we have made a comprehensive detailed numerical calculation by varying the mesh size (grid spacing) in the range from 10 nm to 0.5 μm . In detail, we calculated the free energy of the equilibrium topological states and determine the state diagram dependent on the polarity P_0 and elastic modulus by using these distinct mesh sizes and investigated the difference. Eventually, we find the mesh size in this range will not at all modify the state diagram as shown in Fig. 6. Also, we apologize the insufficient information about free energies of defect cores. In the revised manuscript, the explanations are given for how the energies of line and point defects are evaluated. We also added relevant references. Finally, we found five possible topological states in different conditions of polarization strength and elastic moduli. When changing the mesh size, no shift happens between the state boundaries. For sure, the defect structures for vortex, meron-like variants are unaffected by the mesh size. In the revised manuscript, we replaced the state diagram by the newly-calculated one.

We note that all the original and most of the newly-made simulations were ran from a random initial condition. Disordered polarization states were found to relax to the vortex and meron-variant topologies. For a reference to the Reviewer, we show a dynamic process of the relaxation during the simulation as below (for the D-meron). In the simulations with the revised free-energy, we note that, as shown below, most of the simulations were started from the random disordered state.

However, when the polarization strength is too large and K_{11} is too small, which corresponds to the left corner of the state diagram, a random initial condition will result in somehow disordered result by some trapped defects. For these situations, we use the relaxed structures of the simulation at slightly smaller P_0 or larger K_{11} as the initial condition for the numerical relaxations. We also added explanation about this in the manuscript.

Page 15, Line 11

(Before) “...In the simulation, we create the droplet geometries in Creo Parametric 4.0 and import...Considering the size of droplets observed in our experiment, we set the diameter of the model to be 20 μm and the height to be 5 μm . We perform the relaxation minimization of the corresponding Ginzburg–Landau functionals based on the finite element difference method (within a 3D space of 25 $\mu\text{m} \times 25 \mu\text{m} \times 20 \mu\text{m}$, represented by a $60 \times 60 \times 40$ grid space). Considering, in our experiment, the N_F droplets appear from the disordered Iso state, we set the initial condition of the director field to be random.”

(After) “...The defect core free energy, F_{core} , for three defect-holding polarization states, i.e. simple polarization vortex, escaped polarization vortex and bipolar structure, are calculated as:

$$F_{\text{core}} = \pi K L \ln(R_1/a) \quad (\text{for simple polarization vortex})^{42},$$

$$F_{\text{core}} = 8/3\pi K R_2 \quad (\text{for escaped polarization vortex})^{63},$$

$$F_{\text{core}} = 8\pi K R_2 \quad (\text{for bipolar structure})^{63}.$$

K is taken as the average of the splay, twist and bend elastic moduli. R_1 and R_2 represent the radius of line defects and point defects, respectively. L is the length of line defects. a is the average molecular size. In our calculations, the radius of the line defect is taken to be 0.4 μm and the length is equal to the sample thickness. Under this condition, the calculated energy is comparable to the energy of the line defect in apolar LC systems⁴². Similarly, we assume that the core radius of the point defect in both the bipolar and escaped vortex structures is 0.4 μm . In the simulation, we create the 3D curved cylindrical geometries in Creo Parametric 4.0, which is nearly consistent with the FCPM experimental observation (Extended Figure 3). The 3D models are built to have orthogonal meshes and are used to run the n-director simulations. We set the maximum and minimum diameters of the curved cylinder to be 10 and 5 μm , respectively, which locates in the middle plane of droplets and at the interface between the glass and LCs. The sample thickness is set to be 5 μm . We perform the relaxation minimization of the corresponding Ginzburg–Landau functionals based on the finite element difference method (within a 3D space of 14 $\mu\text{m} \times 14 \mu\text{m} \times 7 \mu\text{m}$). The 3D space is orthogonally divided by grids with the grid spacing in the range from 10 nm to 500 nm. In this range, we make sure that the numerically-calculated topology is consistent, where the topological details do not depend on the size of the grid spacing. Considering, in our experiment, the N_F droplets appear from the disordered Iso state, we set the initial condition of the

director field to be random for most cases. For the conditions at $P_0 < 1 \times 10^{-4} \text{ C m}^{-2}$ and $K_{11}/K_{33} > 0.5$, the simulations end to ideal relaxed topologies as shown in Figure 6g. However, when the polarization strength is large ($P_0 > \sim 1 \times 10^{-4} \text{ C m}^{-2}$) and K_{11} is small ($K_{11}/K_{33} < \sim 0.5$), which corresponds to the left corner of the state diagram, a random initial condition would result in somehow disordered results by some trapped defects. For these situations, we use the relaxed structures of the simulation at slightly smaller P_0 or larger K_{11} as initial condition for the numerical relaxations. ...”

Page23, Figure.6

(Before)

Fig. 6. | Topological structures of the N_F phase in cylindrically confined space. a-c, Order parameter space and director fields of vortex (a), meron (b) and escaped vortex (c). **d,e,** Structural stability diagrams of different material parameters dependent on the ratio of splay modulus and twist modulus (K_{11}/K_{22}), the ratio between dipolar interaction and polarity gradient (β/d), and induced charge effect δ . Blue pentagon, grey rhombus, and orange triangle indicate stable vortex, meron, and escaped-vortex. The shade regions indicate that chirality exist in droplets. Inset schematics show the polarization fields of the xy-plane in the middle of the droplets.

(After)

Fig. 6. | Topological structures of the NF phase in cylindrically confined space. a-e, Director fields and the corresponding order parameter space of simple polarization vortex (a), escaped vortex (b), concentric meron-like structure (c), divergent meron-like structure (d) and bipolar structure (e). **f,** Schematics of the polarization fields on the xy-plane in different depth positions of the droplets. Red points mean point or line defect. **g,** State diagram dependent on the ratio of the splay modulus to the bend modulus (K_{11}/K_{33}) and polarization strength P_0 . The dash and solid lines represent the state boundaries without and with introducing the space charge effect, respectively.

4 In the free energy densities involving \mathbf{P} , do the authors simply replace \mathbf{P} by $P_0\mathbf{n}$ and minimize the free energy functional in terms of \mathbf{n} alone? Then, how is P_0 determined?

-> We apologize for the insufficient description in methods. In the original numerical calculations, we simply replace \mathbf{P} by $P_0\mathbf{n}$ and minimize the free energy functional in terms of \mathbf{n} alone. P_0 is the intrinsic polarity of the polar materials. P_0 is first estimated by a polarization $P_0 = \frac{p}{v}$, where

$p = 13.6$ Debye is the axial molecular dipole moment of RM-OC₂ and v is the volume/molecule in the phase, $v = 325 \text{ cm}^3/\text{mole} = 540 \text{ \AA}^3/\text{molecule}$, assuming a LC mass density of $\rho = 1.3 \text{ g/cm}^3$. Using these parameter values and assuming complete polar ordering of the molecular long axes, we evaluate $P_0 \sim 8.3 \text{ \mu C/cm}^2$, which is close to the experimentally-measured spontaneous polarization value in liquid crystal cells, $\sim 4\text{-}5 \text{ \mu C/cm}^2$.

In the revised manuscript, as we rewrite the free energy equations and only the polarization strength P_0 and elastic moduli as varying parameters for obtaining the state diagram, we simply scan the value of P_0 and elastic moduli for numerical calculation.

5. How are curved boundaries between the isotropic region and the ferroelectric region implemented in the numerical calculations, and what are the boundary conditions there?

-> We apologize for the insufficient description in methods. To implement the curved boundaries, we first create the droplet geometries in Creo Parametric 4.0 and import the STL file into the MATLAB. Using the partial differential equation toolbox, we can convert the 3D model to a 3D geometry with mesh and mapping it to a grid space. This toolbox allows us to find mesh elements and nodes by their geometric location and we obtain the position vectors of the surface nodes. We then use the method described in interpolate the surface nodes and mapping them into orthogonal grids. For the boundary conditions, as described in the original manuscript, we used the degenerate boundary conditions to reduce bound charges ($\sigma_b = \mathbf{P} \cdot \mathbf{v}$, where σ_b is the bound charge density and \mathbf{v} is surface normal vectors). We revised the manuscript as below.

Page 15, Line 11

(Before) "...In the simulation, we create the droplet geometries in Creo Parametric 4.0 and import....Considering the size of droplets observed in our experiment, we set the diameter of the model to be 20 μm and the height to be 5 μm . We perform the relaxation minimization of the corresponding Ginzburg–Landau functionals based on the finite element difference method (within a 3D space of 25 $\mu\text{m} \times 25 \mu\text{m} \times 20 \mu\text{m}$, represented by a 60 \times 60 \times 40 grid space)...."

(After) "...In the simulation, we create the 3D curved cylindrical geometries in Creo Parametric 4.0, which is nearly consistent with the FCPM experimental observation (Extended Figure 3). The 3D models are built to have orthogonal meshes and are used to run the n-director simulations. We set the maximum and minimum diameters of the curved cylinder to be 10 and 5 μm , respectively, which locates in the middle plane of droplets and at the interface between the glass and LCs. The sample thickness is set to be 5 μm . We perform the relaxation minimization of the corresponding Ginzburg–Landau functionals based on the finite element difference method (within a 3D space of 14 $\mu\text{m} \times 14 \mu\text{m} \times 7 \mu\text{m}$)...."

[Characterization of topological structures]

6. In eq. (1), the authors should explicitly state that \mathbf{n} is a unit vector, and what x and y are.

-> We apologize for the insufficient description in the text. In equation (1), the topological number is the integral of solid angle over the whole 2D surface. Directors $\mathbf{n} = \mathbf{n}(x, y)$ are varied with positions so x and y denote the components of position vectors.

Page 2, Line 10

(Before) "...a vectorized quantity, \mathbf{n} , wraps..."

(After) "...a unit vector that defines the average orientation, the so-called director \mathbf{n} , wraps..."

Page 2, Line 13

(Sentence added) " x and y are the Cartesian coordinates."

7. Is Q that first appears three lines after eq. (2) the same as N defined by eq. (1)?

-> We apologize for the confusion. As you suggested, the topological numbers Q is equivalent to N , we revised the manuscript as below.

Page 2, Line 10

(Before) "The Pontryagin number, N , defines how many times a unit vector that defines the average orientation, the so-called director \mathbf{n} , wraps the order parameter space \mathbf{S}^2 (i.e. two-dimensional sphere)^{4, 20}, represented by

$$N = \frac{1}{4\pi} \int \mathbf{n} \cdot (\partial_x \mathbf{n} \times \partial_y \mathbf{n}) dx dy. \quad (1)''$$

(After) "The Pontryagin number, Q , defines how many times a unit vector that defines the average orientation, the so-called director \mathbf{n} , wraps the unit sphere^{4, 20, 21}, represented by

$$Q = \frac{1}{4\pi} \int \mathbf{n} \cdot (\partial_x \mathbf{n} \times \partial_y \mathbf{n}) dx dy. \quad (1)''$$

8. Is φ in the second line of eq. (2) different from ϕ in the first line? If it is, what is φ ? If not, eq. (2) looks very strange.

-> We apologize for the insufficient description in the text. θ and ϕ are polar and azimuthal angles of the director \mathbf{n} while φ is the angle part of the polar coordinate for Cartesian coordinate (x, y) .

Page 2, Line 20

(Before) "..., where θ and ϕ are polar and azimuthal angles. ..."

(After) “..., where θ and ϕ are polar and azimuthal angles of director \mathbf{n} , and φ is the angle part of the polar coordinate for expressing the Cartesian coordinate (x,y)”

9. When the authors first introduce the set $[Q, Q_v, Q_h]$, visualization of examples of structures corresponding to specific sets of $[Q, Q_v, Q_h]$ will help interested readers understand the meaning of these quantities.

-> We thank the Reviewer for this suggestion. We added two examples of meron topology with distinct $[Q, Q_v, Q_h]$ combination that relate to our work as new Figs. 1a,b.

Page 2, Line 20

(Before) “..., where θ and ϕ are polar and azimuthal angles....”

(After) “..., where θ and ϕ are polar and azimuthal angles of director \mathbf{n} , and φ is the angle part of the polar coordinate for expressing the Cartesian coordinate (x,y) . For example, while an anticlockwise Bloch-type meron with an upward vector in the core exhibits a combination of the topological numbers $[Q, Q_v, Q_h] = [1/2, 1, \pi/2]$ (Fig. 1a), a divergent Neel-type anti-meron with a downward vector in the core shows $[Q, Q_v, Q_h] = [-1/2, 1, 0]$ (Fig. 1b).”

Page 18, Caption of Fig. 1

(Sentence added) “**a,b**, Illustrations of 2D merons with different sets of topological charge, vorticity number and helicity number, i.e. $[Q, Q_v, Q_h]$. (a) Bloch-type meron with $[Q, Q_v, Q_h] = [1/2, 1, \pi/2]$. (b) Neel-type anti-meron with $[Q, Q_v, Q_h] = [-1/2, 1, 0]$.”

10. After the authors start discussions on their experimental and numerical results, characterization by $[Q, Q_v, Q_h]$ is made only at the 1st line of page 9. Why do the authors not characterize other structures in the figures by $[Q, Q_v, Q_h]$?

-> We would like to note that in the original manuscript we already characterized the meron-like structures and escape vortex structures by $[Q, Q_v, Q_h]$. In the revision, to further systematically classify topology obtained by the new numerical results, we revised our manuscript and characterized all the structures in the discussion part by $[Q, Q_v, Q_h]$.

Page 8 Line 18

(Sentence added): Please refer to the rewritten Discussion part.

Page 10, Line21

(Sentence added, a part of rewritten Discussion primarily related to the comment): “For C-meron, because the topology is invariant in the depth direction, the topology only carries two possibilities of $[Q, Q_v, Q_h] = [1/2, +1, \pm\pi/2]$, i.e. clockwise or anticlockwise. For both cases, due to the absence of in-plane and out-of-plane twisting structures, they cannot show the non-extinct textures as observed in the experiment (Fig. 3). On the other hand, for D-meron, an in-plane splay-bend deformation and an accompanying spiral streamline of polarization from the center exist. This results in eight species of topology with distinct combinations of topological numbers, $[Q, Q_v, Q_h] = [\pm 1/2, +1, \pm\pi/2]$ and clockwise/counter-clockwise spirals.”

11. In page 9, the authors mention that $Q = 1$ in the “escaped-vortex” structure shown in Fig. 6c. If Q is in fact the same as N defined by eq. (1) (see my previous comment), the integration $\int dx dy$ in one plane would yield $Q = \pm 1/2$. What characterization would yield $Q = 1$?

-> We apologize for this issue. As you mention, the Q should equal to $\pm 1/2$. We revised the related part. Since we rewrote the discussion part on the simulation, please refer to the highlighted manuscript.

Page 10, Line 16

(Sentence added, a part of rewritten Discussion primarily related to the comment): “Introducing polarity into the system dramatically changes the director field. In small K_{11}/K_{33} range, when the polarization strength P_0 increases (e.g., path II in Fig. 6g), due to the preference of antiparallel packing of local polarizations by the enhanced dipolar interaction, the bipolar or escaped vortex director field changes to the defect-less D-meron. In larger K_{11}/K_{33} range ($K_{11}/K_{33} > 0.4 \sim 0.5$), the polar version of C-meron with a pure bend (so concentric) replaces D-meron. For C-meron, because the topology is invariant in the depth direction, the topology only carries two possibilities of $[Q, Q_v, Q_h] = [1/2, +1, \pm\pi/2]$, i.e. clockwise or anticlockwise. For both cases, due to the absence of in-plane and out-of-plane twisting structures, they cannot show the non-extinct textures as observed in the experiment (Fig. 3). On the other hand, for D-meron, an in-plane splay-bend deformation and an accompanying spiral streamline of polarization from the center exist. This results in eight species of topology with distinct combinations of topological numbers, $[Q, Q_v, Q_h] = [\pm 1/2, +1, \pm\pi/2]$ and clockwise/counter-clockwise spirals.”

12. I believe that the gray spheres in Fig. 6 and Extended Data Figures 9 and 10 represent the order parameter space S^2 . In this representation of S^2 , the value of the order parameter is given by the location of a point on the sphere surface (to be more precise, the order parameter is given by a vector pointing to a point on the sphere from the sphere center). Then I am wondering what the distribution of vectors on the sphere surface means. The authors do not seem to understand the distinction between the distribution of the order parameter

in the real space and that in the order parameter space (See, e.g., Figure 12.20 of Kleman and Lavrentovich, “Soft Matter Physics: An introduction”).

-> In the original manuscript, to better to the visibility, we learned from a reference [S.-Z. Lin, A. Saxena, and C. D. Batista, Phys. Rev. B, 91, 224407 (2015)] and drew the order parameter space by the irregular manner. The method provides an intuitive view of the obtained topology on the sphere. On the other hand, we admit the display method is just proposed by some researchers recently and not very widely accepted. Now, we revised the order parameter space as used in the standard method and we revised the Figure 6 as follows.

Page23, Figure.6

(Before)

(After)

13. The “achiral meron” illustrated in Fig. 6d is obviously chiral. In the authors’ “chiral merons” there, splay-bend distortions are involved in the radial directions while they are not in “achiral merons”. This seems to contradict the authors’ explanation in page 8, “This is because filling the space with more ratios of twisting structures costs less elastic energy.” A more convincing argument could be given by close inspection of the free energy densities (eqs. 4-8) and the electrostatic potential φ .

-> We apologize for the confusions. We revised the manuscript as follows. Also, we added many references about the previous reports on apolar texture in nematic droplets, and added discussions about the relationship of these work and present work.

Page 14, Line 6

(Before) “**Numerical Modelling**. As described in the main text, we consider the free energies

composed of the polar gradient (f_p), dipolar interaction (f_d) and the space charge (f_{sc}) contributions besides the Landau (f_L), elastic (f_e) and surface anchoring (f_s) energies:

$$F = \int (f_L + f_e + f_f) dV + \int_{\Omega} f_s d\Omega, \quad (4)$$

$$f_L = \frac{a}{2} |\mathbf{P}|^2 + \frac{b}{2} |\mathbf{P}|^4, \quad (5)$$

$$f_e = \frac{1}{2} K_{11} (\text{div } \mathbf{n})^2 + \frac{1}{2} K_{22} (\mathbf{n} \cdot (\text{curl } \mathbf{n}))^2 + \frac{1}{2} K_{33} (\mathbf{n} \times (\text{curl } \mathbf{n}))^2 - \frac{1}{2} K'_{22} \mathbf{n} \cdot (\nabla \times \mathbf{n}), \quad (6)$$

$$f_f = f_p + f_d + f_{sc} = \frac{d}{2} (\nabla \mathbf{P})^2 + \beta \int \frac{\mathbf{P}(\mathbf{r}) \cdot \mathbf{P}(\mathbf{r}')}{|\mathbf{r} - \mathbf{r}'|^3} d^3r + \delta(\partial_i \varphi) \mathbf{P}_i, \quad (7)$$

$$f_s = \frac{1}{2} W_S (\cos \Theta - \mathbf{n} \cdot \mathbf{v})^2. \quad (8)''$$

(After) “**Numerical Modelling.** As described in the main text, we consider the free energy density contributions from the dipolar interaction (f_d) and the space charge (f_{sc}) contributions besides the Landau (f_L), elastic (f_e), surface anchoring (f_s) and defect core (F_{core}) energies:

$$F = \int (f_L + f_e + f_f) dV + \int_{\Omega} f_s d\Omega + F_{\text{core}}, \quad (4)$$

$$f_L = \frac{a}{2} |\mathbf{P}|^2 + \frac{b}{2} |\mathbf{P}|^4, \quad (5)$$

$$f_e = \frac{1}{2} K_{11} (\text{div } \mathbf{n})^2 + \frac{1}{2} K_{22} (\mathbf{n} \cdot (\text{curl } \mathbf{n}))^2 + \frac{1}{2} K_{33} (\mathbf{n} \times (\text{curl } \mathbf{n}))^2, \quad (6)$$

$$f_d = \frac{1}{4\pi\epsilon_0\epsilon_r} \int \frac{\mathbf{P}(\mathbf{r}') \cdot \mathbf{P}(\mathbf{r})}{|\mathbf{r} - \mathbf{r}'|^3} - \frac{3[\mathbf{P}(\mathbf{r}') \cdot (\mathbf{r} - \mathbf{r}')][\mathbf{P}(\mathbf{r}) \cdot (\mathbf{r} - \mathbf{r}')] }{|\mathbf{r} - \mathbf{r}'|^5} dV, \quad (7)$$

$$f_{sc} = \frac{1}{4\pi\epsilon_0\epsilon_r} \int \frac{\nabla \cdot \mathbf{P}(\mathbf{r}') \cdot \nabla \cdot \mathbf{P}(\mathbf{r})}{|\mathbf{r} - \mathbf{r}'|} dV, \quad (8)$$

$$f_f = f_d + f_{sc}, \quad (9)$$

$$f_s = \frac{1}{2} W_S (\cos \Theta - \mathbf{n} \cdot \mathbf{v})^2. \quad (10)''$$

Page 8, Line 18, Discussion section

(Sentence added): Please refer to the rewritten Discussion part for the comprehensive revisions.

Page 10, Line 6

(Sentence added, a part of rewritten Discussion primarily related to the comment): “We note that, keeping mind that our system has a curved cylindrical shape like a cylinder LC cell, the corresponding apolar textures in the sphere droplet geometry with tangential anchoring have been well studied in traditional nematic materials by simulations and experiments^{55-57, 59-61}. The bipolar structure is easier to form in spherical confined space. It is reported that when $K_{11}/K_{33} \leq 1$, the bipolar structure with two boojums defects can be observed in spherical nematic droplets⁶⁰. Especially, when $K_{11} \geq K_{22} + 0.43K_{33}$, this typical bipolar structure will distort and change to a

twisted bipolar structure⁵⁶. If K_{11} is further increased, the most stable energy structure in the spherical droplet is no longer the apolar bipolar structure, but the apolar concentric structure. This tendency in the reports is consistent with the result for $P_0 = 0$ in our cylindrical droplets.

Introducing polarity into the system dramatically changes the director field. In small K_{11}/K_{33} range, when the polarization strength P_0 increases (e.g., path II in Fig. 6g), due to the preference of antiparallel packing of local polarizations by the enhanced dipolar interaction, the bipolar or escaped vortex director field changes to the defect-less D-meron. In larger K_{11}/K_{33} range ($K_{11}/K_{33} > 0.4\sim 0.5$), the polar version of C-meron with a pure bend (so concentric) replaces D-meron.”

[Other issues]

14. Figures do not appear in numerical order (For example, in page 6, Figs. 6ab are referred to before Figs 4 and 5).

-> We thank your comment. We significantly revised the manuscript and avoided the figures appearing not in numerical order.

15. The first four lines of page 2 exactly match part of the Abstract.

-> We apologize for the issue. We revised the manuscript as below.

Page 2, Line 1

(Before) “The chirality can emerge spontaneously in polar textures and can be biased by introducing chiral dopants. An extended mean-field modelling for the ferroelectric nematic reveals that the stabilization of the polarization meron variants is due to the balance between the elasticity anisotropy and polar interactions like dipolar, polarization gradient, and space charge effect.

(After) “Topological defects arise when symmetry is broken. The complexity and diversity of the resulting spin or magnetic textures, such as skyrmions, merons, and the like, have attracted much attention for centuries¹⁻¹². In recent decades, it has been demonstrated that such vectorized fields with quasi-particle nature serve as an effective approach for integrating and storing magnetic, electronic, optical, and quantum information¹³⁻¹⁹. The findings trigger new challenges for the exploration of the unidentified topological fields, and their designability and controllability.

16. While in eqs. (4-8) the director \mathbf{n} and the polarization \mathbf{P} are treated independently, in page 3 the authors say that the director itself is polar ($\mathbf{n} \neq -\mathbf{n}$). If so, the Frank free energy (6) might need modification to incorporate terms odd in \mathbf{n} . The authors should therefore clarify whether the director is treated as polar, or it is still apolar and the polarity is introduced as \mathbf{P} .

-> In our manuscript, the n-director terms are used to describe the elastic energy, and P is used to

describe the dipole interactions. In page 3, we mentioned: “The N_F state is the polar counterpart of the traditional nematics, which imposes an additional orientational requirement: $\mathbf{n} \neq -\mathbf{n}$ in the order parameter space of S^2 where the head and tail of the director are differentiable (Fig. 1d)”. Also, in the framework of the used n-director model, it is no problem to use the above-mentioned sentence, and say the director \mathbf{n} is polar because the model can differentiate the head and tail of the director, while the Q-tensor model cannot (see the figures as below).

The director field for (b) costs larger energy in the n-director model. Q-tensor model calculates the same free energy for both the structures.

17. Is the “space polarization charge” introduced in page 4 equivalent to N (or Q)?

-> We apologize for the unclear terminology and making the confusion. We revised the manuscript as below.

Page 4, Line 12

(Before) “...In the polar versions, the radial divergent and convergent polarization fields correspond to +1 and -1 space polarization charges (Figs. 1j,k).”

(After) “...In the polar versions, the hedgehog branches into two states that correspond to the radial divergent and convergent polarization fields (Figs. 1l,m).”

18. [Page 5] The word “extinct” is an adjective, not a verb.

-> We apologize for the incorrect grammar. We revised the manuscript as follows. More comprehensive revisions can be found in the revision list and the highlighted revised manuscript.

Page 5, Line 19

(Before) “In N_F droplets, the texture (Figs. 3a-c and Extended Data Fig. 2) does not extinct under the crossed polarizers, especially in the center.”

(After) “In N_F droplets, the texture (Figs. 3a-c and Extended Data Fig. 2) cannot be brought into an extinction status under the crossed polarizers, especially in the center.”

Page 5, Line 23

(Before) “The clockwise (Fig. 3a) or counter-clockwise (Fig. 3c) rotation of the analyzer distinguishes two different types of textures: at specific decrossing conditions, one type extincts and the other remains bright.”

(After) “The clockwise (Fig. 3a) or counter-clockwise (Fig. 3c) rotation of the analyzer distinguishes two different types of textures: at specific decrossing conditions, one type turns to a dark state (i.e. extinction) and the other remains bright.”

19. Are SHG-CPM and SHG-I developed by the authors for the first time for this work? If not, references on them should be cited.

-> For SHG-I measurement, the system is built as in Ref. [33, 36 and 54], which we have cited in original manuscript. We additionally added a new reference [53] for SHG-CPM.

20. Should I_n be I_k in eq. (3)?

-> We apologize for the confusion in the symbolic abbreviation. We replaced I_n by I_k in equation (3) and replace a_n by a_k .

Page 13, Line 2

(Before) “...The detected fluorescence intensity follows the equation:

$$I_n = I_{back} + I_{norm} \cos^4(\psi - k\pi/4) \cos^4 \theta. \quad (3)$$

k takes a range from 0 to 3 and the angle between polarization directions and x axis is $k\pi/4$. I_{back} is the background intensity and I_{norm} the normalized fluorescence. I_n is the transmittance at the polarization condition of α_n .”

(After) “...The detected fluorescence intensity follows the equation:

$$I_k = I_{back} + I_{norm} \cos^4(\psi - k\pi/4) \cos^4 \theta. \quad (3)$$

k takes a range from 0 to 3 and the angle between polarization directions and x axis is $k\pi/4$. I_{back} is the background intensity and I_{norm} the normalized fluorescence. I_k is the transmittance at the polarization condition of α_k .”

21. The phrase “short-range antiferroelectric-like ordering” in page 12 might confuse the readers because the dipolar interaction f_d itself is long-ranged.

-> We apologize for the misleading description. In our new numerical model, we rewrote the dipolar interaction terms, $f_d = \frac{1}{4\pi\epsilon_0\epsilon_r} \int \frac{\mathbf{P}(\mathbf{r}') \cdot \mathbf{P}(\mathbf{r})}{|\mathbf{r} - \mathbf{r}'|^3} - \frac{3[\mathbf{P}(\mathbf{r}') \cdot (\mathbf{r} - \mathbf{r}')][\mathbf{P}(\mathbf{r}) \cdot (\mathbf{r} - \mathbf{r}')]}{|\mathbf{r} - \mathbf{r}'|^5} dV$. We made a more detailed description in the revised manuscript. For many details, please refer to the rewritten Method part.

Page 14, Line 21

(Before) “...the dipolar interaction, f_d , prefers the short-range antiferroelectric-like ordering.”

(After) “...The dipolar interaction, f_d , describes the electrostatic contributions from all the interacting polar bodies in the system, which includes the boundary charge due to the discontinuity

of the polarization on the interface. When a polarization field in the bulk is inhomogeneous, it creates ‘virtual’ charges (called space charges). ...”

22. Visibility of the chemical formula and “ $\mu = 13.5D$ ” in Fig. 1b is quite poor.

-> To improve the visibility, we updated the appearance of the chemical formula and “ $\mu = 13.6 D$ ” for clearer presentations as below.

Page 18, Figure 1

23. The arrows in Extended Data Figures 9 and 10 are hard to distinguish visually.

-> Thank you for your suggestion. For better visibility and clarity, we redraw the figures and added the related figures as below (Extended Data Fig. 9-10 after revision).

Extended Data Fig. 9 | Comparison of simulation results between concentric meron-like structure and divergent meron-like structure in confined space. a,b, The XZ-cross-sectional polarization field of concentric meron-like structure (a) and divergent meron-like structure (b).

Extended Data Fig. 10 | Numerical simulation results of possible divergent meron-like structure states and their corresponding anti-meron-like structures in confined space. a-h, Four types of polarization fields of divergent meron-like structure classified by the combination of handedness of the spirals and polarization orientation. **i-p,** Four types of polarization fields of the divergent (convergent) anti-meron-like structures classified by the combination of their handedness of the spirals handedness and polarization orientation.

To Reviewer 3:

Thank you very much for reviewing our manuscript and giving us your positive comments. We highly appreciate your invaluable and helpful feedbacks to our manuscript. Based on the Reviewer's comments, we made additional experiments on probing the spontaneous polarization and ferroelectricity of the band-like texture by the decrossing technique and FCPM observation. We answered your questions and significantly revised our manuscript as stated in the following according to your comments and suggestions. A comprehensive revision list can be found at the end of the reply. Your positive decision would be very much appreciated.

1) The molecule, RM-OC₂, is newly synthesized, and the liquid crystal phase of this molecule should be thoroughly discussed in this article. During the cooling, there might be three phase-transition points (67.1°C, 108.4°C, and 131.4°C). What are their phases in each temperature range? There is a minimal discussion about the phase in this article.

-> We apologize for the insufficient information. We added a full phase sequence on both cooling and heating as below.

Page 5, Line 1

(Before) "...RM-OC₂ exhibits a direct transition from the isotropic liquid (Iso) to the N_F state upon cooling (Figs. 2a-d, Extended Data Figs. 1b-j)."

(After) "...RM-OC₂ exhibits a direct transition from the isotropic liquid (Iso) to the N_F state upon cooling with the phase sequence of Iso-[64.6 °C]-N_F (Figs. 2a-d). On the heating, the material shows several recrystallization processes, i.e. N_F-[67.1 °C]-Iso-[82.1 °C]-unknown crystal X1-[105-120 °C]-unknown crystal X2-[140.1 °C]-Iso on heating (Extended Data Figs. 1b-j). The recrystallization processes suggest the Iso phase is metastable with respect to some crystalline forms in the range of 80-150 °C."

2) As a follow-up to question 1, there is an unusual phase transition in Extended Data Fig. 1f-g. After the textures become clear at 69 °C, which might be the isotropic phase, even though the temperature increase, new nucleation occurs. This is a very unusual phenomenon in liquid crystal materials, and the authors should account for it.

-> This is a typical phenomenon for undercooled matter states. For example, the undercooled water prevents the system from crystallization, but a small shock or local temperature rise will cause the water crystallized. In the present case, at low temperatures on cooling, though crystal is the most stable state, the slow nucleation and growth kinetics does not allow the system crystallized at the observation time scale (up to days). However, we can observe very small non-growing nucleus of crystal phase. Upon heating, the thermal perturbation provides fluctuation to the system to transit to the most-stable crystal phase. As shown in Extended Data Figs. 1f-g, the domains readily crystallized

upon heating can be attributed to the seeding effect from the remaining nucleus. This is also clearly observed in the DSC measurement (Extended Data Fig. 1).

Page 21, **Supplementary Discussion 1**

(Before) “The phase behaviors of RM-OC₂ are shown in Figures 2b-d and Extended Data Figures 1c-j. The phase behavior upon the heating process is not simply mirror of that of the cooling process. During the temperature increases from 63 °C to 150 °C, the band-like texture of N_F phase gradually melts (Extended Data Figs. 1c-f) and some crystal nuclei emerge. The crystalline then disappears upon further heating (Extended Data Figs. 1g-j).”

(After) “The phase behaviors of RM-OC₂ are shown in Figures 2b-d and Extended Data Figures 1c-j. The phase behavior upon the heating process is not simply mirror of that of the cooling process, exhibiting the phase sequence of N_F-[67.1 °C]-Iso-[82.1 °C]-unknown crystal X1-[105-120 °C]-unknown crystal X2-[140.1 °C]-Iso. During the temperature increases from 63 °C to 150 °C, the band-like texture of N_F phase gradually melts into the Iso phase (Extended Data Figs. 1c-f), and some crystal nuclei emerge. The crystalline phase X1 is replaced by another crystalline phase X2 at high temperatures, and then disappears upon further heating at 140.1 °C (Extended Data Figs. 1g-j). The transition process suggests the metastability of the system. At low temperatures on cooling, though crystal is the most stable state, the slow nucleation and growth kinetics does not allow the system crystallized at the observation time scale (up to days). However, we can observe very small non-growing nucleus of crystal phase. Upon heating, the thermal perturbation provides fluctuation to the system to transit to the most-stable crystal phase. As shown in Extended Data Figs. 1f-g, the domains readily crystallized upon heating can be attributed to the seeding effect from the remaining nucleus. This is also clearly observed in the DSC measurement (Extended Data Fig. 1).”

3) In Figure 2d, the textures are distinct from the conventional nematic, which indicates that the head-to-tail symmetry of the director is spontaneously broken. The author investigated the paper's 3D director and polarization of meron variants. It is necessary to perform 3D analysis (confocal & SHG) and PLM analysis (with de-crossing polarizers) even in the state shown in Figure 2d to discuss more details for the N_F phase.

-> Thank you for your suggestion. To clarify director and polarization fields of the band-like texture in Fig. 2d, we added to the PLM, FCPM, SHG-CPM images as shown in the Extended Data Figs. 12 and 13. Also, we explained the band and stripes in a more detailed way in Supplementary Discussion 4.

Page 23, **Supplementary Discussion 4**

(Sentence Added) **Supplementary Discussion 4.** According to our observation, RM-OC₂ directly transits from isotropic to N_F with an intermittent co-existence (Figs. 2b,c). When the sample is further cooled down, the droplets merge and transit to a band-like texture, where the line disclinations run mainly along the rubbing direction (Fig. 2d). The corresponding director and polarization field can

be directly visualized by SHG microscopy. Extended Data Fig. S15a demonstrates a large-area 2D SHG confocal polarizing microscopy (SHG-CPM) image of the band texture for the cell midplane during the cooling at 30 °C. In each domain, the SH signal shows the maximum intensity when the polarization is parallel to rubbing direction. This means the polarization points along the rubbing direction. Similar to Ref. (1) [Li, J., Nishikawa, H., Kougo, J., *et al. Sci. Adv.* 2021, 7(17): eabf5047.], the SHG interferometry reveals that the neighboring domains exhibit opposite polarity.

4) In Figure 3l, m, the authors drew the vector field in the cylinder. However, the author discussed that the meron variants deviated from cylindrical shapes. Please check it.

-> We apologize for the confusion caused by Figures 3l-m. The purpose of using cylinder shape in Figure 3l, m was to show the director more concisely and clearly. However, the shape of the droplet used in calculations is 3D curved cylindrical shape as revealed by confocal microscopy. The size of the used cylinder is equal to the geometry and the size of droplet observed in FCPM (Extended Figure 3). The XY-cross section of the cylinder on the interface between glass and N_F LC is a circular shape with radius of 5 μm . The XY-cross section in the middle plane is a circular shape with radius of 10 μm . The sample thickness is 10 μm . To state more clearly, we revised the droplet shape in Figures 3l-m and described the used shape geometry for calculations more detailed.

Page 15, Line 11

(Before) "...In the simulation, we create the droplet geometries in Creo Parametric 4.0 and import...Considering the size of droplets observed in our experiment, we set the diameter of the model to be 20 μm and the height to be 5 μm . We perform the relaxation minimization of the corresponding Ginzburg–Landau functionals based on the finite element difference method (within a 3D space of 25 μm ×25 μm ×20 μm , represented by a 60×60×40 grid space). Considering, in our experiment, the N_F droplets appear from the disordered Iso state, we set the initial condition of the director field to be random."

(After) "...In the simulation, we create the 3D curved cylindrical geometries in Creo Parametric 4.0, which is nearly consistent with the FCPM experimental observation (Extended Figure 3). The 3D models are built to have orthogonal meshes and are used to run the n-director simulations. We set the maximum and minimum diameters of the curved cylinder to be 10 and 5 μm , respectively, which locates in the middle plane of droplets and at the interface between the glass and LCs. The sample thickness is set to be 5 μm . We perform the relaxation minimization of the corresponding Ginzburg–Landau functionals based on the finite element difference method (within a 3D space of 14 μm × 14 μm × 7 μm). ..."

Page 20, Figures 3l,m

(Before)

l

m

(After)

l

m

5) The vector fields in Extended Data Fig. 9 and Extended Data Fig. 10 are too dense to be seen. Please reduce the number of them and edit them to make them more visible.

-> Thank you for your suggestion. For better visibility and clarity, we redraw the original Extended Data Figs. 9 and 10 and added the related figures as below (Extended Data Figs. 9-10 in revised version).

Extended Data Fig. 9 | Comparison of simulation results between concentric meron-like structure and divergent meron-like structure in confined space. a,b, The XZ-cross-sectional polarization field of concentric meron-like structure (a) and divergent meron-like structure (b).

Extended Data Fig. 10 | Numerical simulation results of possible divergent meron-like structure states and their corresponding anti-meron-like structures in confined space. a-h, Four types of polarization fields of divergent meron-like structure classified by the combination of handedness of the spirals and polarization orientation. **i-p,** Four types of polarization fields of the divergent (convergent) anti-meron-like structures classified by the combination of their handedness of the spirals handedness and polarization orientation.

List of Major Revisions

1. We revised all the terms “merons” to “meron-like structures”.

2. We revised the term “achiral merons” to “concentric meron-like structures” and “chiral merons” to “divergent (convergent) meron-like structures” in our new numerical results.

3. Page 2, Line 1

(Before) “The chirality can emerge spontaneously in polar textures and can be biased by introducing chiral dopants. An extended mean-field modelling for the ferroelectric nematic reveals that the stabilization of the polarization meron variants is due to the balance between the elasticity anisotropy and polar interactions like dipolar, polarization gradient, and space charge effect.

(After) “Topological defects arise when symmetry is broken. The complexity and diversity of the resulting spin or magnetic textures, such as skyrmions, merons, and the like, have attracted much attention for centuries¹⁻¹². In recent decades, it has been demonstrated that such vectorized fields with quasi-particle nature serve as an effective approach for integrating and storing magnetic, electronic, optical, and quantum information¹³⁻¹⁹. The findings trigger new challenges for the exploration of the unidentified topological fields, and their designability and controllability.

4. Page 2 Line 10

(Before) “The Pontryagin number, N , defines how many times a unit vector that defines the average orientation, the so-called director \mathbf{n} , wraps the order parameter space \mathbf{S}^2 (i.e. two-dimensional sphere)^{4, 20}, represented by

$$N = \frac{1}{4\pi} \int \mathbf{n} \cdot (\partial_x \mathbf{n} \times \partial_y \mathbf{n}) dx dy. \quad (1)''$$

(After) “The Pontryagin number, Q , defines how many times a unit vector that defines the average orientation, the so-called director \mathbf{n} , wraps the order parameter space \mathbf{S}^2 (i.e. two-dimensional sphere)^{4, 20, 21}, represented by

$$Q = \frac{1}{4\pi} \int \mathbf{n} \cdot (\partial_x \mathbf{n} \times \partial_y \mathbf{n}) dx dy. \quad (1)''$$

5. Page 2 Line 20

(Before) “..., where θ and ϕ are polar and azimuthal angles. ...”

(After) “..., where θ and ϕ are polar and azimuthal angles of director \mathbf{n} , and φ is the angle part of the polar coordinate for expressing the Cartesian coordinate (x,y)”

6. Page 4 Line 12

(Before) “...In the polar versions, the radial divergent and convergent polarization fields correspond to +1 and -1 space polarization charges (Figs. 1j,k).”

(After) "...In the polar versions, the hedgehog branches into two states that correspond to the radial divergent and convergent polarization fields (Figs. 1l,m)."

7. Page 5 Line 1

(Before) "...RM-OC₂ exhibits a direct transition from the isotropic liquid (Iso) to the N_F state upon cooling (Figs. 2a-d, Extended Data Figs. 1b-j)."

(After) "...RM-OC₂ exhibits a direct transition from the isotropic liquid (Iso) to the N_F state upon cooling with the phase sequence of Iso-[64.6 °C]-N_F (Figs. 2a-d). On the heating, the material shows several recrystallization processes, i.e. N_F-[67.1 °C]-Iso-[82.1 °C]-unknown crystal X1-[105-120 °C]-unknown crystal X2-[140.1 °C]-Iso on heating (Extended Data Figs. 1b-j). The recrystallization processes suggest the Iso phase is metastable with respect to some crystalline forms in the range of 80-150 °C."

8. Page 5, Line 19

(Before) "**Observation of polarization meron variants.** In N_F droplets, the texture (Figs. 3a-c and Extended Data Fig. 2) does not extinct under the crossed polarizers, especially in the center."

(After) "**Observation of polarization meron variants.** In N_F droplets, the texture (Figs. 3a-c and Extended Data Fig. 2) can not be brought into an extinction appearance under the crossed polarizers, especially in the center."

9. Page 5, Line 23

(Before) "The clockwise (Fig. 3a) or counter-clockwise (Fig. 3c) rotation of the analyzer distinguishes two different types of textures: at specific decrossing conditions, one type extincts and the other remains bright."

(After) "The clockwise (Fig. 3a) or counter-clockwise (Fig. 3c) rotation of the analyzer distinguishes two different types of textures: at specific decrossing conditions, one type turns a dark texture appearance (extinction) and the other remains bright."

10. Page 8, Line 18, Discussion section

(Sentence added): "Please refer to the rewritten Discussion part for the comprehensive revisions."

11. Page 13, Line 1

(Before) "...The detected fluorescence intensity follows the equation:

$$I_n = I_{back} + I_{norm} \cos^4(\psi - k\pi/4) \cos^4 \theta. \quad (3)$$

k takes a range from 0 to 3 and the angle between polarization directions and x axis is $k\pi/4$. I_{back} is the background intensity and I_{norm} the normalized fluorescence. I_n is the transmittance at the polarization condition of α_n ."

(After) "...The detected fluorescence intensity follows the equation:

$$I_k = I_{back} + I_{norm} \cos^4(\psi - k\pi/4) \cos^4 \theta. \quad (3)$$

k takes a range from 0 to 3 and the angle between polarization directions and x axis is $k\pi/4$. I_{back} is the background intensity and I_{norm} the normalized fluorescence. I_k is the transmittance at the polarization

condition of α_k .”

12. Page 14 Line 6

(Before) “**Numerical Modelling.** As described in the main text, we consider the free energies composed of the polar gradient (f_p), dipolar interaction (f_d) and the space charge (f_{sc}) contributions besides the Landau (f_L), elastic (f_e) and surface anchoring (f_s) energies:

$$F = \int (f_L + f_e + f_f) dV + \int_{\Omega} f_s d\Omega, \quad (4)$$

$$f_L = \frac{a}{2} |\mathbf{P}|^2 + \frac{b}{2} |\mathbf{P}|^4, \quad (5)$$

$$f_e = \frac{1}{2} K_{11} (\text{div } \mathbf{n})^2 + \frac{1}{2} K_{22} (\mathbf{n} \cdot (\text{curl } \mathbf{n}))^2 + \frac{1}{2} K_{33} (\mathbf{n} \times (\text{curl } \mathbf{n}))^2 - \frac{1}{2} K'_{22} \mathbf{n} \cdot (\nabla \times \mathbf{n}), \quad (6)$$

$$f_f = f_p + f_d + f_{sc} = \frac{d}{2} (\nabla \mathbf{P})^2 + \beta \int \frac{\mathbf{P}(\mathbf{r}) \cdot \mathbf{P}(\mathbf{r}')}{|\mathbf{r} - \mathbf{r}'|^3} d^3 r + \delta(\partial_i \varphi) \mathbf{P}_i, \quad (7)$$

$$f_s = \frac{1}{2} W_S (\cos \Theta - \mathbf{n} \cdot \mathbf{v})^2. \quad (8)''$$

(After) “**Numerical Modelling.** As described in the main text, we consider the free energy density contributions from the dipolar interaction (f_d) and the space charge (f_{sc}) contributions besides the Landau (f_L), elastic (f_e), surface anchoring (f_s) and defect core (F_{core}) energies:

$$F = \int (f_L + f_e + f_f) dV + \int_{\Omega} f_s d\Omega + F_{\text{core}}, \quad (4)$$

$$f_L = \frac{a}{2} |\mathbf{P}|^2 + \frac{b}{2} |\mathbf{P}|^4, \quad (5)$$

$$f_e = \frac{1}{2} K_{11} (\text{div } \mathbf{n})^2 + \frac{1}{2} K_{22} (\mathbf{n} \cdot (\text{curl } \mathbf{n}))^2 + \frac{1}{2} K_{33} (\mathbf{n} \times (\text{curl } \mathbf{n}))^2, \quad (6)$$

$$f_d = \frac{1}{4\pi\epsilon_0\epsilon_r} \int \frac{\mathbf{P}(\mathbf{r}') \cdot \mathbf{P}(\mathbf{r})}{|\mathbf{r} - \mathbf{r}'|^3} - \frac{3[\mathbf{P}(\mathbf{r}') \cdot (\mathbf{r} - \mathbf{r}')][\mathbf{P}(\mathbf{r}) \cdot (\mathbf{r} - \mathbf{r}')] }{|\mathbf{r} - \mathbf{r}'|^5} dV, \quad (7)$$

$$f_{sc} = \frac{1}{4\pi\epsilon_0\epsilon_r} \int \frac{\nabla \cdot \mathbf{P}(\mathbf{r}') \cdot \nabla \cdot \mathbf{P}(\mathbf{r})}{|\mathbf{r} - \mathbf{r}'|} dV, \quad (8)$$

$$f_f = f_d + f_{sc}, \quad (9)$$

$$f_s = \frac{1}{2} W_S (\cos \Theta - \mathbf{n} \cdot \mathbf{v})^2. \quad (10)''$$

13. Page 14, Line 18

Sentence added: “ P_0 is the intrinsic polarization strength of polar molecules.”

14. Page 14, Line 20

(Sentence added) “The free energy term, f_f , is the total polar terms including the dipolar interaction, f_d , and space charge effect, f_{sc} . The dipolar interaction, f_d , describes the electrostatic contributions from all the interacting polar bodies in the system, which includes the boundary charge due to the discontinuity of the polarization on the interface. When a polarization field in the bulk is inhomogeneous, it creates ‘virtual’

charges (called space charge). Especially, the splay polarization field $\nabla \cdot \mathbf{P}(\mathbf{r})$ is the most energetically unfavorable. To include the space charge effect, f_{sc} , we calculate the electrostatic contributions from all the interacting space charges in the bulk⁶². ϵ_0 is the vacuum permittivity, $8.85 \times 10^{-12} \text{ C V}^{-1} \text{ m}^{-1}$, ϵ_r is the dielectric constant of materials. Here, due to a constant of ϵ_r is used for calculation, though the contribution of the variation of the polarization field is considered, the orientation-dependent dielectric interaction is not fully accounted in the present form.”

15. Page 15, Line 11

Sentence added: “The defect core free energy, F_{core} , for three defect-holding polarization states, i.e. simple polarization vortex, escaped polarization vortex and bipolar structure, are calculated as:

$$F_{\text{core}} = \pi KL \ln(R_1/a) \text{ (for simple polarization vortex)}^{42},$$

$$F_{\text{core}} = 8/3\pi KR_2 \text{ (for escaped polarization vortex)}^{63},$$

$$F_{\text{core}} = 8\pi KR_2 \text{ (for bipolar structure)}^{63}.$$

K is taken as the average of the splay, twist and bend elastic moduli. R_1 and R_2 represent the radius of line defects and point defects, respectively. L is the length of line defects. a is the average molecular size. In our calculations, the radius of the line defect is taken to be $0.4 \mu\text{m}$ and the length is equal to the sample thickness. Under this condition, the calculated energy is comparable to the energy of the line defect in apolar LC systems⁴². Similarly, we assume that the core radius of the point defect in both the bipolar and escaped vortex structures is $0.4 \mu\text{m}$.”

16. Page 15, Line 22

(Before) “...In the simulation, we create the droplet geometries in Creo Parametric 4.0 and import....Considering the size of droplets observed in our experiment, we set the diameter of the model to be $20 \mu\text{m}$ and the height to be $5 \mu\text{m}$. We perform the relaxation minimization of the corresponding Ginzburg–Landau functionals based on the finite element difference method (within a 3D space of $25 \mu\text{m} \times 25 \mu\text{m} \times 20 \mu\text{m}$, represented by a $60 \times 60 \times 40$ grid space). Considering, in our experiment, the N_F droplets appear from the disordered Iso state, we set the initial condition of the director field to be random.”

(After) “...In the simulation, we create the 3D curved cylindrical geometries in Creo Parametric 4.0, which is nearly consistent with the FCPM experimental observation (Extended Figure 3). The 3D models are built to have orthogonal meshes and are used to run the n-director simulations. We set the maximum and minimum diameters of the curved cylinder to be 10 and $5 \mu\text{m}$, respectively, which locates in the middle plane of droplets and at the interface between the glass and LCs. The sample thickness is set to be $5 \mu\text{m}$. We perform the relaxation minimization of the corresponding Ginzburg–Landau functionals based on the finite element difference method (within a 3D space of $14 \mu\text{m} \times 14 \mu\text{m} \times 7 \mu\text{m}$). The 3D space is orthogonally divided by grids with the grid spacing in the range from 10 nm to 500 nm . In this range, we make sure that the numerically-calculated topology is consistent, where the topological details do not depend on the size of the grid spacing. Considering, in our experiment, the N_F droplets appear from the disordered Iso state, we set the initial condition of the director field to be random for most cases. For the conditions at $P_0 < 1 \times 10^{-4} \text{ C}$

m^{-2} and $K_{11}/K_{33} > 0.5$, the simulations end to ideal relaxed topologies as shown in Figure 6g. However, when the polarization strength is large ($P_0 > \sim 1 \times 10^{-4} \text{ C m}^{-2}$) and K_{11} is small ($K_{11}/K_{33} < \sim 0.5$), which corresponds to the left corner of the state diagram, a random initial condition would result in somehow disordered results by some trapped defects. For these situations, we use the relaxed structures of the simulation at slightly smaller P_0 or larger K_{11} as initial condition for the numerical relaxations....”

17. Page 18, Caption of Fig. 1

(Sentence added) “**a,b**, Illustrations of 2D merons with different sets of topological charge, vorticity number and helicity number, i.e. $[Q, Q_v, Q_h]$. (a) Bloch-type meron with $[Q, Q_v, Q_h] = [1/2, 1, \pi/2]$. (b) Neel-type anti-meron with $[Q, Q_v, Q_h] = [-1/2, 1, 0]$.”

18. Page 19, Caption of Fig. 2

(Before) “...Bottom chart shows the temperature dependencies of the dielectric constant (blue squares). ...”

(After) “... Bottom chart shows the temperature dependencies of the dielectric constant at a frequency of 25 Hz (blue squares). ...”

19. Page 21, Caption of Fig. 4

(Before) “**p-s**, Reconstructed polarization fields with left hand inward polarization directions (p), left hand outward polarization directions (q), right hand outward polarization directions (r) and right hand inward polarization directions (s).”

(After) “**p-s**, Reconstructed polarization fields with right-handed convergent polarization field (p), left-handed divergent polarization field (q), right-handed divergent polarization field (r) and left-handed convergent polarization field (s).”

20. We revised the Figures 1 and 3.

21. We revised the Figure 6. Also, we revised the captions.

22. We revised the Extended Data Figures 9 and 10. Also, we revised their captions.

23. We added new Extended Data Figures 11, 12, 13, 14 and 15.

24. We revised **Supplementary Discussion 1** in Supplementary Materials.

(Before) “The phase behaviors of RM-OC₂ are shown in Figures 2b-d and Extended Data Figures 1c-j. The phase behavior upon the heating process is not simply mirror of that of the cooling process. During the temperature increases from 63 °C to 150 °C, the band-like texture of N_F phase gradually melts (Extended Data Figs. 1c-f) and some crystal nuclei emerge. The crystalline then disappears upon further heating (Extended Data Figs. 1g-j).”

(After) “The phase behaviors of RM-OC₂ are shown in Figures 2b-d and Extended Data Figures 1c-j. The

phase behavior upon the heating process is not simply mirror of that of the cooling process, exhibiting the phase sequence of N_F -[67.1 °C]-Iso-[82.1 °C]-unknown crystal X1-[105-120 °C]- unknown crystal X2-[140.1 °C]-Iso. During the temperature increases from 63 °C to 150 °C, the band-like texture of N_F phase gradually melts into the Iso phase (Extended Data Figs. 1c-f), and some crystal nuclei emerge. The crystalline phase X1 is replaced by another crystalline phase X2 at high temperatures, and then disappears upon further heating at 140.1 °C (Extended Data Figs. 1g-j). The transition process suggests the metastability of the system. At low temperatures on cooling, though crystal is the most stable state, the slow nucleation and growth kinetics does not allow the system crystallized at the observation time scale (up to days). However, we can observe very small non-growing nucleus of crystal phase. Upon heating, the thermal perturbation provides fluctuation to the system to transit to the most-stable crystal phase. As shown in Extended Data Figs. 1f-g, the domains readily crystallized upon heating can be attributed to the seeding effect from the remaining nucleus. This is also clearly observed in the DSC measurement (Extended Data Fig. 1).”

25. We added new **Supplementary Discussion 4** in Supplementary Materials.

(Sentence Added) **Supplementary Discussion 4.** According to our observation, RM-OC₂ directly transits from isotropic to N_F with an intermittent co-existence (Figs. 2b,c). When the sample is further cooled down, the droplets merge and transit to a band-like texture, where the line disclinations run mainly along the rubbing direction (Fig. 2d). The corresponding director and polarization field can be directly visualized by SHG microscopy. Extended Data Fig. S15a demonstrates a large-area 2D SHG confocal polarizing microscopy (SHG-CPM) image of the band texture for the cell midplane during the cooling at 30 °C. In each domain, the SH signal shows the maximum intensity when the polarization is parallel to rubbing direction. This means the polarization points along the rubbing direction. Similar to Ref. (1) [Li, J., Nishikawa, H., Kougo, J., *et al. Sci. Adv.* 2021, 7(17): eabf5047.], the SHG interferometry reveals that the neighboring domains exhibit opposite polarity.

26. We rewrote the Discussion part.

(Sentence Added) Therefore, we consider an extended Frank-Oseen free energy function of the system by incorporating dipolar interactions and additional space charge energies besides the Landau, elastic, surface anchoring and defect core energies (Methods). In general, irrespective of the effect of the space charge, we uncover five possible polarization states in the curved cylindrically confined space: polarization vortex (Fig. 6a), escaped vortex (Fig. 6b), concentric meron-like structure (C-meron; Fig. 6c), divergent meron-like structures (D-meron, including the convergent, so negatively divergent, meron-like structure; Fig. 6d) and bipolar structure (Fig. 6e). In the simplest polarization vortex (Fig. 6a), the polarization field arranges in a pure concentric manner and lies on the sample plane (i.e. wrapping the equator of the order parameter space S^2 sphere). This leads to a disclination line in the center. In the escaped vortex (Fig. 6b), the polarization continuously flips along the depth direction, enabling the disclination line of simple polarization vortex to escape onto the middle plane. Now, the polarizations span over the whole S^2 sphere. Since the disclination line of the simple polarization vortex costs much larger energy than that of the point defect of the escaped vortex in our experimental geometry (see Methods for free energy descriptions), the escaped vortex is always

stable over the simple polarization vortex in the simulation conditions (Fig. 6g and Extended Data Figs. 11-13). Meron-like structures are distinct from the vortices because they do not exhibit singularities. C-meron demonstrates a pure concentric polarization field except for the core region (Figs. 6c). D-meron exhibits additional directional splay-bend distortions (Figs. 6d), which is consistent with the experimental observation of the in-plane handedness of meron-like structures. The bipolar structure possesses two boojum defects on the north and south poles (Figs. 6e), which has been well studied in the classical apolar nematic droplets⁵⁵⁻⁵⁸. Details of the two-dimensional distribution of polarization fields at different sample depth for all the structures is shown in Fig. 6f.

It is important to mention that the experimentally-observed D-meron like structure appears only when K_{11} satisfies proper conditions: $0.2 \text{ pN} < K_{11} < 1 \text{ pN}$. The variation of K_{22} or K_{33} with respect to other elastic moduli seems to have less potential to induce D-meron (Extended Data Figs. 11-13). Herein, we demonstrate a representative simulated state diagram by varying the effective polarization strength P_0 and the elastic anisotropy (Fig. 6g). The elastic anisotropy is defined as a ratio of the splay to bend elasticity K_{11}/K_{33} (assuming the twist and bend elastic modulus are 0.8 pN and 0.2 pN , respectively, as are closed to the reported values in the vicinity of the N-N_F phase transition⁴⁷). Both results without (dashed line state boundaries) and with (solid line state boundaries) the space charge effect are shown. We note that the space charge effect only shifts the state boundaries but does not essentially change the state diagram. Therefore, we mainly discuss the results without the space charge term in the following. When the polarity is absent in the system (so $P_0 = 0$), the bipolar structure with a large portion of splay deformation is dominant in small K_{11}/K_{33} values ($K_{11}/K_{33} < 0.4$). With increasing K_{11}/K_{33} , so increasing the penalty of the splay deformation^{42, 55, 56, 59}, the bipolar structure becomes unstable and is replaced by the escaped vortex-like apolar director field. Further increasing the K_{11}/K_{33} will cause a further transition to an apolar variant of the C-meron. We note that, keeping mind that our system has a curved cylindrical shape like a cylinder LC cell, the corresponding apolar textures in the sphere droplet geometry with tangential anchoring have been well studied in traditional nematic materials by simulations and experiments^{55-57, 59-61}. The bipolar structure is easier to form in spherical confined space. It is reported that when $K_{11}/K_{33} \leq 1$, the bipolar structure with two boojums defects can be observed in spherical nematic droplets⁶⁰. Especially, when $K_{11} \geq K_{22} + 0.43K_{33}$, this typical bipolar structure will distort and change to a twisted bipolar structure⁵⁶. If K_{11} is further increased, the most stable energy structure in the spherical droplet is no longer the apolar bipolar structure, but the apolar concentric structure. This tendency in the reports is consistent with the result for $P_0 = 0$ in our cylindrical droplets.

Introducing polarity into the system dramatically changes the director field. In small K_{11}/K_{33} range, when the polarization strength P_0 increases (e.g., path II in Fig. 6g), due to the preference of antiparallel packing of local polarizations by the enhanced dipolar interaction, the bipolar or escaped vortex director field changes to the defect-less D-meron. In larger K_{11}/K_{33} range ($K_{11}/K_{33} > 0.4 \sim 0.5$), the polar version of C-meron with a pure bend (so concentric) replaces D-meron. For C-meron, because the topology is invariant in the depth direction, the topology only carries two possibilities of $[Q, Q_v, Q_h] = [1/2, +1, \pm\pi/2]$, i.e. clockwise or anticlockwise. For both cases, due to the absence of in-plane and out-of-plane twisting structures, they cannot show the non-extinct textures as observed in the experiment (Fig. 3). On the other hand, for D-meron,

an in-plane splay-bend deformation and an accompanying spiral streamline of polarization from the center exist. This results in eight species of topology with distinct combinations of topological numbers, $[Q, Q_v, Q_h] = [\pm 1/2, +1, \pm \pi/2]$ and clockwise/counter-clockwise spirals. However, the polarization fields of a meron and an anti-meron with opposite handedness are degenerate in experiments since the current nonlinear optical microscopy have no axial resolution for differentiating the polarization states along light propagation direction, i.e. up or down along the viewing direction (Extended Data Fig. 9-10). Thus, only four types of meron-like structures are experimentally observed (Fig. 4I-o). The appearance necessitates the system possessing a strong polarization strength over the transition threshold, i.e. $P_0 > 10^{-4} \text{ C m}^{-2}$.

Finally, let us comment on the effect of the space charge effect. Though it is not essential in modifying the state diagram, it has a role to shift the state boundaries (from solid to dashed boundaries; Fig. 6g). It is seen that, if the space charge effect is dominant, the experimentally-observed D-meron would be destabilized because of its effect to expel the splay deformation. Therefore, we expect the space charge effect might not be very influential in the current system. This would be due to the partial existence of ionic species for charge screening in the synthesized materials. We note that, though the current calculation of the space charge term considers the contribution from the variation of the polarization field that dislikes the splay deformation, the isotropic distribution of the dielectric permittivity in space is considered through using a dielectric constant of ϵ_r . The exploration of the full description of the anisotropic electrostatic interactions in polar liquid crystals is beyond the current scope and will be explored in future. However, considering the role of space charge revealed above, we expect the full introduction of the anisotropic dielectric interaction may only work to slightly shift the phase boundaries.

Reviewers' comments:

Reviewer #1 (Remarks to the Author):

In the revised manuscript »Spontaneous electric-polarization topology in confined ferroelectric nematics «, the authors responded to most of the points raised by the reviewers adequately, however, the electrostatic energy is still not correctly expressed. So before I can recommend the paper for publication in Nat Comm this issue must be amended.

As I already wrote in my first review, the electrostatic self-energy can be included in two different ways, either as the interaction between dipoles or by using bound charges. Only one approach should be used, not both. Eq. (7) describes the dipolar energy and, if that is used, there is no other electrostatic term, so the fsc (Eq.(8)) term should be omitted because Eq(7) already includes the inhomogeneity of the polarization. If the calculation with bound charges is chosen, then the term in Eq. (7) should be omitted. If the authors chose to use a bound charge description, then in (Eq.(8)) the contribution of the surface term is missing. If the authors chose to use dipolar description, then in Eq (7), a factor $\frac{1}{2}$ is missing as each pair of the dipoles is counted twice.

Reviewer #2 (Remarks to the Author):

See the attached PDF file.

Reviewer #3 (Remarks to the Author):

The manuscript has been extensively well-revised to be accepted for publication.

The authors made substantial revisions in response to the three reviewers including me. However, there are still serious problems in the manuscript and some of the authors' responses are not satisfactory, as described below. Therefore I do not recommend the publication of this version.

- The authors do not understand the comment (1) of Reviewer 1 on electrostatics of dipole moments. The authors' f_d (eq. 7) is the dipole-dipole interaction energy (although the factor $1/2$ is missing) in the “microscopic” picture in the nomenclature of Reviewer 1, and f_{sc} (eq. 8) is that in the “macroscopic picture”. Namely, f_d and f_{sc} are the **same** dipolar interaction energy from different standpoints. Reviewer 1 clearly says, “So there are not two different contributions (a dipolar one and the one from the bound charges), they are the same because the bound charges come from the dipoles.” Then the arguments on the space charge (pages 9-11) do not make sense.

I also comment that the authors should make clear with respect to what the nablas (∇) are in (eq. 8), and with respect to what the integration dV is carried out (eqs. 4,7,8).

- In my first comment, I said, “The value of the parameters in the free energy densities (eqs. 4-8) are not at all given.” They are still not given. What are elastic constants? What are the values of a , b , ϵ_r and W_S ?
- Regarding my fourth comment, I understand that f_L (eq. 5) dictates the magnitude of the polarization P_0 , but in their reply, the authors state that P_0 is a varying parameter and that they simply scan the value of P_0 . It is strange.
- The authors do not appropriately answer my 16th comment on the polarity of \mathbf{n} . If \mathbf{n} is polar, the Frank elastic energy should comprise not only the familiar three terms (splay, bend and twist), but also terms that are odd in \mathbf{n} and therefore not allowed in an apolar nematic (for example, $(\mathbf{n} \cdot \nabla \times \mathbf{n}) \nabla \cdot \mathbf{n}$). Nevertheless, the authors' numerical calculations are based on the Frank elastic energy containing only the three terms (eq. 6).
- [page 15] What is the “radius of line defects and point defects”? I also note that “core radius” usually means the characteristic size of a defect core determined by the nematic correlation length ($\simeq 10$ nm).

- [Fig. 6f] Readers outside the field of liquid crystals may not be familiar with the nail symbol.
- Several minor typographical errors:
 - [page 9, 4th line from the bottom] closed → close
 - [page 16, 8th line] Howere → However(?)

Point-to-Point Reply to the Reviewers

In the following, the statement of the reviewer and our answers/discussions appear in *black* and *blue* fonts, respectively.

To Reviewer 1:

Thank you very much for reviewing our manuscript and we highly appreciate your invaluable and helpful feedbacks to our manuscript. Based on the Reviewer's comment, we revised the free energy functional and redraw the state diagram. The comprehensive revision can be found in the highlighted manuscript. We hope that you are satisfied with our replies and revision. Your positive decision would be very much appreciated.

1) As I already wrote in my first review, the electrostatic self-energy can be included in two different ways, either as the interaction between dipoles or by using bound charges. Only one approach should be used, not both. Eq. (7) describes the dipolar energy and, if that is used, there is no other electrostatic term, so the f_{sc} (Eq.(8)) term should be omitted because Eq(7) already includes the inhomogeneity of the polarization. If the calculation with bound charges is chosen, then the term in Eq. (7) should be omitted. If the authors chose to use a bound charge description, then in (Eq.(8)) the contribution of the surface term is missing. If the authors chose to use dipolar description, then in Eq (7), a factor $\frac{1}{2}$ is missing as each pair of the dipoles is counted twice.

-> Thank you for your comments, and we must apologize for the previous inadequacy on the description of the polar terms. As suggested, we chose to use the dipolar energy for the electrostatic self-energy description, and removed the additional space charge term. We comprehensively revised the corresponding part. In Eq (7), a factor of 1/2 is also included as kindly suggested by the reviewer. Now the state diagram is updated, which is essentially same to the previous one with some shifts of the state boundaries. We also slightly revised the manuscript to explain the result.

Page 9, Line 17

(Before) "It is important to mention that the experimentally-observed D-meron like structure appears only when K_{11} satisfies proper conditions: $0.2 \text{ pN} < K_{11} < 1.0 \text{ pN}$."

(After) "It is important to mention that the experimentally-observed D-meron like structure appears only when K_{11} satisfies proper conditions: $0.2 \text{ pN} < K_{11} < 0.8 \text{ pN}$."

Page 9, Line 24

(Before) "When the polarity is absent in the system (so $P_0 = 0$), the bipolar structure with a large portion of splay deformation is dominant in small K_{11}/K_{33} values ($K_{11}/K_{33} < 0.4\sim 0.5$)."

(After) "When the polarity is absent or small in the system (so $P_0 < 2.5 \times 10^{-4} \text{ C m}^{-2}$), the bipolar structure with a large portion of splay deformation is dominant in small K_{11}/K_{33} values ($K_{11}/K_{33} < 0.4$)."

Page 11, Line 1

(Before) “The appearance necessitates the system possessing a strong polarization strength over the transition threshold, i.e. $P_0 > 10^{-4} \text{ C m}^{-2}$ ”

(After) “The appearance necessitates the system possessing a strong polarization strength over the transition threshold, i.e. $P_0 > 2.5 \times 10^{-4} \text{ C m}^{-2}$.”

Page 13, Line 15

(Before) “**Numerical Modelling.** As described in the main text, we consider the free energy density contributions from the dipolar interaction (f_d) and the space charge (f_{sc}) contributions besides the Landau (f_L), elastic (f_e), surface anchoring (f_s) and defect core (F_{core}) energies:

$$F = \int (f_L + f_e + f_f) dV + \int_{\Omega} f_s d\Omega + F_{\text{core}} , \quad (4)$$

$$f_L = \frac{a}{2} |\mathbf{P}|^2 + \frac{b}{2} |\mathbf{P}|^4, \quad (5)$$

$$f_e = \frac{1}{2} K_{11} (\text{div } \mathbf{n})^2 + \frac{1}{2} K_{22} (\mathbf{n} \cdot (\text{curl } \mathbf{n}))^2 + \frac{1}{2} K_{33} (\mathbf{n} \times (\text{curl } \mathbf{n}))^2, \quad (6)$$

$$f_d = \frac{1}{4\pi\epsilon_0\epsilon_r} \int \frac{\mathbf{P}(\mathbf{r}') \cdot \mathbf{P}(\mathbf{r})}{|\mathbf{r} - \mathbf{r}'|^3} - \frac{3[\mathbf{P}(\mathbf{r}') \cdot (\mathbf{r} - \mathbf{r}')][\mathbf{P}(\mathbf{r}) \cdot (\mathbf{r} - \mathbf{r}')]}{|\mathbf{r} - \mathbf{r}'|^5} dV, \quad (7)$$

$$f_{sc} = \frac{1}{4\pi\epsilon_0\epsilon_r} \int \frac{\nabla \cdot \mathbf{P}(\mathbf{r}') \cdot \nabla \cdot \mathbf{P}(\mathbf{r})}{|\mathbf{r} - \mathbf{r}'|} dV, \quad (8)$$

$$f_f = f_d + f_{sc} , \quad (9)$$

$$f_s = \frac{1}{2} W_S (\cos \theta - \mathbf{n} \cdot \mathbf{v})^2. \quad (10)''$$

(After) “**Numerical Modelling.** As described in the main text, we consider the free energy density contributions from the dipolar interaction (f_d) besides the Landau (f_L), elastic (f_e), surface anchoring (f_s) and defect core (F_{core}) energies:

$$F = \int (f_L + f_e + f_d) dV + \int_{\Omega} f_s d\Omega + F_{\text{core}} , \quad (4)$$

$$f_L = \frac{a}{2} |\mathbf{P}|^2 + \frac{b}{2} |\mathbf{P}|^4, \quad (5)$$

$$f_e = \frac{1}{2} K_{11} (\text{div } \mathbf{n})^2 + \frac{1}{2} K_{22} (\mathbf{n} \cdot (\text{curl } \mathbf{n}))^2 + \frac{1}{2} K_{33} (\mathbf{n} \times (\text{curl } \mathbf{n}))^2, \quad (6)$$

$$f_d = \frac{1}{8\pi\epsilon_0\epsilon_r} \int \left\{ \frac{\mathbf{P}(\mathbf{r}') \cdot \mathbf{P}(\mathbf{r})}{|\mathbf{r} - \mathbf{r}'|^3} - \frac{3[\mathbf{P}(\mathbf{r}') \cdot (\mathbf{r} - \mathbf{r}')][\mathbf{P}(\mathbf{r}) \cdot (\mathbf{r} - \mathbf{r}')]}{|\mathbf{r} - \mathbf{r}'|^5} \right\} dV, \quad (7)$$

$$f_s = \frac{1}{2} W_S (\cos \theta - \mathbf{n} \cdot \mathbf{v})^2. \quad (8)''$$

Page 14, Line 4

(Before) “The dipolar interaction, f_d , describes the electrostatic contributions from all the interacting polar bodies in the system, which includes the boundary charge due to the discontinuity of the polarization on the interface.”

(After) “The dipolar interaction, f_d , describes the electrostatic contributions from all the interacting polar bodies in the system, which includes both the space charge arising from the splay polarization field, $\nabla \cdot \mathbf{P}(\mathbf{r})$ ⁶³, and boundary charge due to the discontinuity of the polarization on the interface.”

Page 23, Figure 6

(Before)

(After)

Extended Data Figure 11 in Supplementary Materials

(Before)

(After)

Extended Data Figure 12 in Supplementary Materials

(Before)

(After)

Extended Data Figure 13 in Supplementary Materials

(Before)

(After)

To Reviewer 2:

Thank you very much for reviewing our manuscript and we highly appreciate your invaluable and helpful feedbacks to our manuscript. Based on the Reviewer's comment, we revised the free energy functional and redraw the state diagram. The point-by-point reply to the Reviewer's comments is shown below. The comprehensive revision can be found in the highlighted manuscript. We hope that you are satisfied with our replies and revision. Your positive decision would be very much appreciated.

1) The authors do not understand the comment (1) of Reviewer 1 on electrostatics of dipole moments. The authors' f_d (eq. 7) is the dipole-dipole interaction energy (although the factor 1/2 is missing) in the "microscopic" picture in the nomenclature of Reviewer 1, and f_{sc} (eq. 8) is that in the "macroscopic picture". Namely, f_d and f_{sc} are the same dipolar interaction energy from different standpoints. Reviewer 1 clearly says, "So there are not two different contributions (a dipolar one and the one from the bound charges), they are the same because the bound charges come from the dipoles." Then the arguments on the space charge (pages 9-11) do not make sense.

We thank for your valuable comments. Following the reviewer #1's and your suggestions, we made the corresponding revisions. Please refer to our response to reviewer #1. Now the state diagram is updated, which is essentially same to the previous one with some shifts of the state boundaries. We deleted the unnecessary parts and revised the manuscript to explain the result. Please also refer to the highlighted manuscript.

Page 9, Line 17

(Before) "It is important to mention that the experimentally-observed D-meron like structure appears only when K_{11} satisfies proper conditions: $0.2 \text{ pN} < K_{11} < 1.0 \text{ pN}$."

(After) "It is important to mention that the experimentally-observed D-meron like structure appears only when K_{11} satisfies proper conditions: $0.2 \text{ pN} < K_{11} < 0.8 \text{ pN}$."

Page 9, Line 24

(Before) "When the polarity is absent in the system (so $P_0 = 0$), the bipolar structure with a large portion of splay deformation is dominant in small K_{11}/K_{33} values ($K_{11}/K_{33} < 0.4\sim 0.5$)."

(After) "When the polarity is absent or small in the system (so $P_0 < 2.5 \times 10^{-4} \text{ C m}^{-2}$), the bipolar structure with a large portion of splay deformation is dominant in small K_{11}/K_{33} values ($K_{11}/K_{33} < 0.4$)."

Page 11, Line 1

(Before) "The appearance necessitates the system possessing a strong polarization strength over the transition threshold, i.e. $P_0 > 10^{-4} \text{ C m}^{-2}$ "

(After) "The appearance necessitates the system possessing a strong polarization strength over the transition threshold, i.e. $P_0 > 2.5 \times 10^{-4} \text{ C m}^{-2}$."

Page 13, Line 15

(Before) “**Numerical Modelling.** As described in the main text, we consider the free energy density contributions from the dipolar interaction (f_d) and the space charge (f_{sc}) contributions besides the Landau (f_L), elastic (f_e), surface anchoring (f_s) and defect core (F_{core}) energies:

$$F = \int (f_L + f_e + f_f) dV + \int_{\Omega} f_s d\Omega + F_{core} , \quad (4)$$

$$f_L = \frac{a}{2} |\mathbf{P}|^2 + \frac{b}{2} |\mathbf{P}|^4, \quad (5)$$

$$f_e = \frac{1}{2} K_{11} (\text{div } \mathbf{n})^2 + \frac{1}{2} K_{22} (\mathbf{n} \cdot (\text{curl } \mathbf{n}))^2 + \frac{1}{2} K_{33} (\mathbf{n} \times (\text{curl } \mathbf{n}))^2, \quad (6)$$

$$f_d = \frac{1}{4\pi\epsilon_0\epsilon_r} \int \frac{\mathbf{P}(\mathbf{r}') \cdot \mathbf{P}(\mathbf{r})}{|\mathbf{r}-\mathbf{r}'|^3} - \frac{3[\mathbf{P}(\mathbf{r}') \cdot (\mathbf{r}-\mathbf{r}')][\mathbf{P}(\mathbf{r}) \cdot (\mathbf{r}-\mathbf{r}')] }{|\mathbf{r}-\mathbf{r}'|^5} dV, \quad (7)$$

$$f_{sc} = \frac{1}{4\pi\epsilon_0\epsilon_r} \int \frac{\nabla \cdot \mathbf{P}(\mathbf{r}') \cdot \nabla \cdot \mathbf{P}(\mathbf{r})}{|\mathbf{r}-\mathbf{r}'|} dV, \quad (8)$$

$$f_f = f_d + f_{sc} , \quad (9)$$

$$f_s = \frac{1}{2} W_S (\cos \theta - \mathbf{n} \cdot \mathbf{v})^2. \quad (10)''$$

(After) “**Numerical Modelling.** As described in the main text, we consider the free energy density contributions from the dipolar interaction (f_d) and the space charge (f_{sc}) contributions besides the Landau (f_L), elastic (f_e), surface anchoring (f_s) and defect core (F_{core}) energies:

$$F = \int (f_L + f_e + f_d) dV + \int_{\Omega} f_s d\Omega + F_{core} , \quad (4)$$

$$f_L = \frac{a}{2} |\mathbf{P}|^2 + \frac{b}{2} |\mathbf{P}|^4, \quad (5)$$

$$f_e = \frac{1}{2} K_{11} (\text{div } \mathbf{n})^2 + \frac{1}{2} K_{22} (\mathbf{n} \cdot (\text{curl } \mathbf{n}))^2 + \frac{1}{2} K_{33} (\mathbf{n} \times (\text{curl } \mathbf{n}))^2, \quad (6)$$

$$f_d = \frac{1}{8\pi\epsilon_0\epsilon_r} \int \left\{ \frac{\mathbf{P}(\mathbf{r}') \cdot \mathbf{P}(\mathbf{r})}{|\mathbf{r}-\mathbf{r}'|^3} - \frac{3[\mathbf{P}(\mathbf{r}') \cdot (\mathbf{r}-\mathbf{r}')][\mathbf{P}(\mathbf{r}) \cdot (\mathbf{r}-\mathbf{r}')] }{|\mathbf{r}-\mathbf{r}'|^5} \right\} dV, \quad (7)$$

$$f_s = \frac{1}{2} W_S (\cos \theta - \mathbf{n} \cdot \mathbf{v})^2. \quad (8)''$$

Page 14, Line 4

(Before) “The dipolar interaction, f_d , describes the electrostatic contributions from all the interacting polar bodies in the system, which includes the boundary charge due to the discontinuity of the polarization on the interface.”

(After) “The dipolar interaction, f_d , describes the electrostatic contributions from all the interacting polar bodies in the system, which includes both the space charge arising from the splay polarization field, $\nabla \cdot \mathbf{P}(\mathbf{r})$ ⁶³, and boundary charge due to the discontinuity of the polarization on the interface.”

(Before)

(After)

Extended Data Figure 11 in Supplementary Materials

(Before)

(After)

Extended Data Figure 12 in Supplementary Materials

(Before)

(After)

Extended Data Figure 13 in Supplementary Materials

(Before)

(After)

2) I also comment that the authors should make clear with respect to what the nablas (∇) are in (eq. 8), and with respect to what the integration dV is carried out (eqs. 4,7,8).

-> In our new revision, the surface charge term of (eq. 8) with the nabla (∇) is now removed. Since we are using a finite element simulation method, the droplet is meshed into many finite small volume elements, i.e. $dV = dx dy dz$. We numerically integrated the corresponding free energy density function in eq. (4) using dV . we added some descriptions as shown below.

Page 14, Line 18

(Sentence added) "According to eq. (4), the free energy terms are integrated using the finite volume and surface area elements $dV = dx dy dz$ and $d\Omega = dx dy$, respectively."

3) In my first comment, I said, "The value of the parameters in the free energy densities (eqs. 4-8) are not at all given." They are still not given. What are elastic constants? What are the values of a , b , ϵ_r and W_S ?

-> We apologize for missing values in some physical terms. The description of elastic constants was in the original manuscript: "The elastic anisotropy is defined as a ratio of the splay to bend elasticity K_{11}/K_{33} (assuming the twist and bend elastic modulus are 0.8 pN and 2.0 pN, respectively, as are close to the reported values in the vicinity of the N-N $_F$ phase transition⁴⁷).". We added the used physical parameters (ϵ_r and W_S) into the manuscript. Though the Landau term is useful for determining whether a system would choose a finite P_0 to realize the polar phase under a certain combination of (a , b), it does not crucial for determining a topological state at a constant P_0 (when a polar phase is already assumed). We also supplemented some explanations for this in the new revision.

Page 14, Line 8

(Sentence added) “ ϵ_r is the relative permittivity, which is set to be 10^4 in our simulations. The value is comparable to the value reported in the literatures^{27, 48}.”

Page 14, Line 16

(Sentence added) “To reasonably consider this, we employ the degenerate planar anchoring condition⁶⁴, i.e. $\theta = \pi/2$ and $W_s = 10^{-6} \text{ J/m}^2$.”

Page 15, Line 13

(Sentence added) “To investigate how the strength of polarity and elasticities would affect the topological nature, we vary P_0 and the ratio of elasticities, K_{11}/K_{33} , for the simulations. Note that, unlike the simulation by using the Landau–de Gennes Q-tensor⁴² where the order parameter can change during orientational relaxation, the predesignated P_0 for each simulation condition in the present free-energy context is unchanged during simulation runs since both the polarity strength (so the order parameter of polarization) and the nematic order parameter are constants, i.e. $|\mathbf{P}| = P_0$ and $|\mathbf{n}| = 1$. Under this circumstance, the Landau energy in eq. (5) is a constant for a chosen combination of $(P_0, K_{11}/K_{33})$. This means that the term will not affect what topological states would be chosen. Therefore, in our simulation, the Landau term is discarded.”

4) Regarding my fourth comment, I understand that f_L (eq. 5) dictates the magnitude of the polarization P_0 , but in their reply, the authors state that P_0 is a varying parameter and that they simply scan the value of P_0 . It is strange.

-> In a simulation for each condition of P_0 and elasticity in the state diagram, different values of P_0 and elasticity ratios are provided for the numerical calculation. Please also refer to the reply in (3). To supplement the explanation on this part, we also added some descriptions as shown below.

Page 15, Line 13

(Sentence added) “To investigate how the strength of polarity and elasticities would affect the topological nature, we vary P_0 and the ratio of elasticities, K_{11}/K_{33} , for the simulations. Note that, unlike the simulation by using the Landau–de Gennes Q-tensor⁴² where the order parameter can change during orientational relaxation, the predesignated P_0 for each simulation condition in the present free-energy context is unchanged during simulation runs since both the polarity strength (so the order parameter of polarization) and the nematic order parameter are constants, i.e. $|\mathbf{P}| = P_0$ and $|\mathbf{n}| = 1$. Under this circumstance, the Landau energy in eq. (5) is a constant for a chosen combination of $(P_0, K_{11}/K_{33})$. This means that the term will not affect what topological states would be chosen. Therefore, in our simulation, the Landau term is discarded.”

5) The authors do not appropriately answer my 16th comment on the polarity of \mathbf{n} . If \mathbf{n} is polar, the Frank elastic energy should comprise not only the familiar three terms (splay, bend and twist), but also terms that

are odd in n and therefore not allowed in an apolar nematic (for example, $(\mathbf{n} \cdot \nabla \times \mathbf{n})\nabla \cdot \mathbf{n}$). Nevertheless, the authors' numerical calculations are based on the Frank elastic energy containing only the three terms (eq. 6).

-> As we described before, we consider that the polar interaction only arises in the polar terms. Therefore, we used the traditional nematic elastic potentials which were also used for describing ferroelectric or polar nematics previously by several groups [49,62]. Of course, if one considers that the elastic interactions are with its origin of polarity, it is necessary to change the description of the terms as suggested by the Reviewer. This approach will be a future work in the field. We slightly revised the manuscript as below.

Page 14, Line 1

(Sentence added) "The elastic energy term, f_e , treats the traditional nematic elastic functionals, where only the terms that are quadratic in \mathbf{n} are used^{49,62}. It..."

6) [page 15] What is the "radius of line defects and point defects"? I also note that "core radius" usually means the characteristic size of a defect core determined by the nematic correlation length (≈ 10 nm).

-> Based on the above-mentioned revisions on the simulation, we supplemented the information on the energy of the defect core and revised the manuscript as below.

Page 14, Line 2 from the bottom

(Before) "In our calculations, the radius of the line defect is taken to be $0.4 \mu\text{m}$ and the length is equal to the sample thickness. Under this condition, the calculated energy is comparable to the energy of the line defect in apolar LC systems⁴². Similarly, we assume that the core radius of the point defect in both the bipolar and escaped vortex structures is $0.4 \mu\text{m}$."

(After) "The polar vortex contains a line defect (Fig. 6a), whose length is equal to the sample thickness. Under this condition, the calculated energy is comparable to the energy of the line defect reported in apolar LC systems⁴². The escaped vortex and bipolar structures carry point defects (Fig. 6b and Fig. 6e). The core radius of the defects is assumed to be comparable to the nematic correlation length, i.e. $R_1 = R_2 = 15 \text{ nm}$ ⁶⁵."

(Before)

(After)

Extended Data Figure 11 in Supplementary Materials

(Before)

(After)

Extended Data Figure 12 in Supplementary Materials

(Before)

(After)

Extended Data Figure 13 in Supplementary Materials

7) [Fig. 6f] Readers outside the field of liquid crystals may not be familiar with the nail symbol. Several minor typographical errors:

- [page 9, 4th line from the bottom] closed \rightarrow close
- [page 16, 8th line] Howere \rightarrow However(?)

\rightarrow We added the explanation of the nail symbol in the caption of Fig. 6.

We apologize for the typos. All of these are corrected.

Page 23, Caption of Fig. 6

(Before) "... f, Schematics of the polarization fields on the xy-plane in different depth positions of the droplets. ..."

(After) "... f, The polarization fields projected on the xy-plane in different depth positions of the droplets are described nail vectors. ..."

Page 9, Line 5 from the bottom

(Before) "... as are closed to the reported values in the vicinity of the N-N_F phase transition⁴⁷..."

(After) "... as are close to the reported values in the vicinity of the N-N_F phase transition⁴⁷..."

Page 15, Line 3 from the bottom

(Before) "... Howere, ..."

(After) "... However, ..."

REVIEWERS' COMMENTS

Reviewer #1 (Remarks to the Author):

In the 2nd revision of the manuscript »Spontaneous electric-polarization topology in confined ferroelectric nematics «, the authors responded to most of the points raised by the reviewers adequately, and I recommend the paper for publication in Nat Comm.